# BrainSCUBA: Fine-Grained Natural Language Captions of Visual Cortex Selectivity

**Andrew F. Luo**
Carnegie Mellon University
afluo@cmu.edu

**Margaret M. Henderson**
Carnegie Mellon University
mmhender@cmu.edu

**Michael J. Tarr**
Carnegie Mellon University
michaeltarr@cmu.edu

**Leila Wehbe**
Carnegie Mellon University
lwehbe@cmu.edu

## Abstract

Understanding the functional organization of higher visual cortex is a central focus in neuroscience. Past studies have primarily mapped the visual and semantic selectivity of neural populations using hand-selected stimuli, which may potentially bias results towards pre-existing hypotheses of visual cortex functionality. Moving beyond conventional approaches, we introduce a data-driven method that generates natural language descriptions for images predicted to maximally activate individual voxels of interest. Our method – Semantic Captioning Using Brain Alignments ("BrainSCUBA") – builds upon the rich embedding space learned by a contrastive vision-language model and utilizes a pre-trained large language model to generate interpretable captions. We validate our method through fine-grained voxel-level captioning across higher-order visual regions. We further perform text-conditioned image synthesis with the captions, and show that our images are semantically coherent and yield high predicted activations. Finally, to demonstrate how our method enables scientific discovery, we perform exploratory investigations on the distribution of "person" representations in the brain, and discover fine-grained semantic selectivity in body-selective areas. Unlike earlier studies that decode text, our method derives *voxel-wise captions of semantic selectivity*. Our results show that BrainSCUBA is a promising means for understanding functional preferences in the brain, and provides motivation for further hypothesis-driven investigation of visual cortex. Code and project site: https://www.cs.cmu.edu/~afluo/BrainSCUBA

## 1 Introduction

The recognition of complex objects and semantic visual concepts is supported by a network of regions within higher visual cortex. Past research has identified the specialization of certain regions in processing semantic categories such as faces, places, bodies, words, and food (Puce et al., 1996; Kanwisher et al., 1997; McCarthy et al., 1997; Maguire, 2001; Epstein & Kanwisher, 1998; Grill-Spector, 2003; Downing et al., 2001; Khosla et al., 2022; Pennock et al., 2023; Jain et al., 2023). Notably, the discovery of these regions has largely relied on a hypothesis-driven approach, whereby the researcher hand-selects stimuli to study a specific hypothesis. This approach risk biasing the results as it may fail to capture the complexity and variability inherent in real-world images, which can lead to disagreements regarding a region's functional selectivity (Gauthier et al., 1999).

To better address these issues, we introduce **BrainSCUBA** (Semantic Captioning Using Brain Alignments), an approach for synthesizing *per-voxel* natural language captions that describe voxel-wise *preferred stimuli*. Our method builds upon the availability of large-scale fMRI datasets (Allen et al., 2022) with a natural image viewing task, and allows us to leverage contrastive vision-language models and large-language models in identifying fine-grained voxel-wise functional specialization in a data-driven manner. BrainSCUBA is conditioned on weights from an image-computable fMRI encoder that maps from image to voxel-wise brain activations. The design of our encoder allows us to extract the optimal encoder embedding for each voxel, and we use a training-free method to close the modality gap between the encoder-weight space and natural images. The output of BrainSCUBA

describes (in words) the visual stimulus that maximally activates a given voxel. Interpretation and visualization of these captions facilitates data-driven investigation into the underlying feature preferences across various visual sub-regions in the brain.

In contrast to earlier studies that decode text from the brain activity related to an image, we demonstrate *voxel-wise captions* of semantic selectivity. Concretely, we show that our method captures the categorical selectivity of multiple regions in visual cortex. Critically, the content of the captions replicates the field's pre-existing knowledge of each region's preferred category. We further show that BrainSCUBA combined with a text-to-image model can generate images semantically aligned with targeted brain regions and yield high predicted activations when evaluated with a different encoder backbone. Finally, we use BrainSCUBA to perform data-driven exploration for the coding of the category "person", finding evidence for person-selective regions outside of the commonly recognized face/body domains and discovering new finer-grained selectivity within known body-selective areas.

## 2 RELATED WORK

Several recent studies have yielded intriguing results by using large-scale vision-language models to reconstruct images and text-descriptions from brain patterns when viewing images (Takagi & Nishimoto, 2022; Chen et al., 2022; Doerig et al., 2022; Ferrante et al., 2023; Ozcelik & VanRullen, 2023; Liu et al., 2023), or to generate novel images that are predicted to activate a given region (Ratan Murty et al., 2021; Gu et al., 2022; Luo et al., 2023). Broadly speaking, these approaches require conditioning on broad regions of the visual cortex, and have not demonstrated the ability to scale down and enable voxel-level understanding of neural selectivity. Additionally, these methods produce images rather than interpretable captions. Work on artificial neurons (Borowski et al., 2020; Zimmermann et al., 2021) have shown that feature visualization may not be more informative than top images in artificial neural networks. In contrast, our work tackles biological networks which have more noisy top-images that are less conducive to direct analysis, and the synthesis of novel images/captions can act as a source of stimuli for future hypothesis-driven neuroscience studies.

**Semantic Selectivity in Higher Visual Cortex.** Higher visual cortex in the human brain contains regions which respond selectively to specific categories of visual stimuli, such as faces, places, bodies, words, and food (Desimone et al., 1984; Puce et al., 1996; Kanwisher et al., 1997; McCarthy et al., 1997; Maguire, 2001; Epstein & Kanwisher, 1998; Grill-Spector, 2003; Downing et al., 2001; Cohen et al., 2000; Khosla et al., 2022; Pennock et al., 2023; Jain et al., 2023). These discoveries have predominantly relied on the use of hand-selected stimuli designed to trigger responses of distinct regions. However the handcrafted nature of these stimuli may misrepresent the complexity and diversity of visual information encountered in natural settings (Gallant et al., 1998; Felsen & Dan, 2005). In contrast, the recent progress in fMRI encoders that map from stimulus to brain response have enabled data-driven computational tests of brain selectivity in vision (Naselaris et al., 2011; Huth et al., 2012; Yamins et al., 2014; Eickenberg et al., 2017; Wen et al., 2018; Kubilius et al., 2019; Conwell et al., 2023; Wang et al., 2022), language (Huth et al., 2016; Deniz et al., 2019), and at the interface of vision and language (Popham et al., 2021). Here, based on Conwell et al. (2023)'s evaluation of the brain alignment of various pre-trained image models, we employ CLIP as our encoder backbone.

**Image-Captioning with CLIP and Language Models.** Vision-language models trained with a contrastive loss demonstrate remarkable capability across many discriminative tasks (Radford et al., 2021; Cherti et al., 2023; Sun et al., 2023). However, due to the lack of a text-decoder, these models are typically paired with an adapted language model in order to produce captions. When captioning, some models utilize the full spatial CLIP embedding (Shen et al., 2021; Li et al., 2023a), whilst others use only the vector embedding (Mokady et al., 2021; Tewel et al., 2022; Li et al., 2023b). By leveraging the multi-modal latent space learned by CLIP, we are able to generate voxel-wise captions without human-annotated voxel-caption data.

**Brain-Conditioned Image and Caption Generation.** There are two broad directions when it comes to brain conditioned generative models for vision. The first seeks to decode (reconstruct) visual inputs from the corresponding brain activations, including works that leverage retrieval, variational autoencoders (VAEs), generative adversarial networks (GANs), and score/energy/diffusion models (Kamitani & Tong, 2005; Han et al., 2019; Seeliger et al., 2018; Shen et al., 2019; Ren et al., 2021; Takagi & Nishimoto, 2022; Chen et al., 2023; Lu et al., 2023; Ozcelik & VanRullen, 2023).

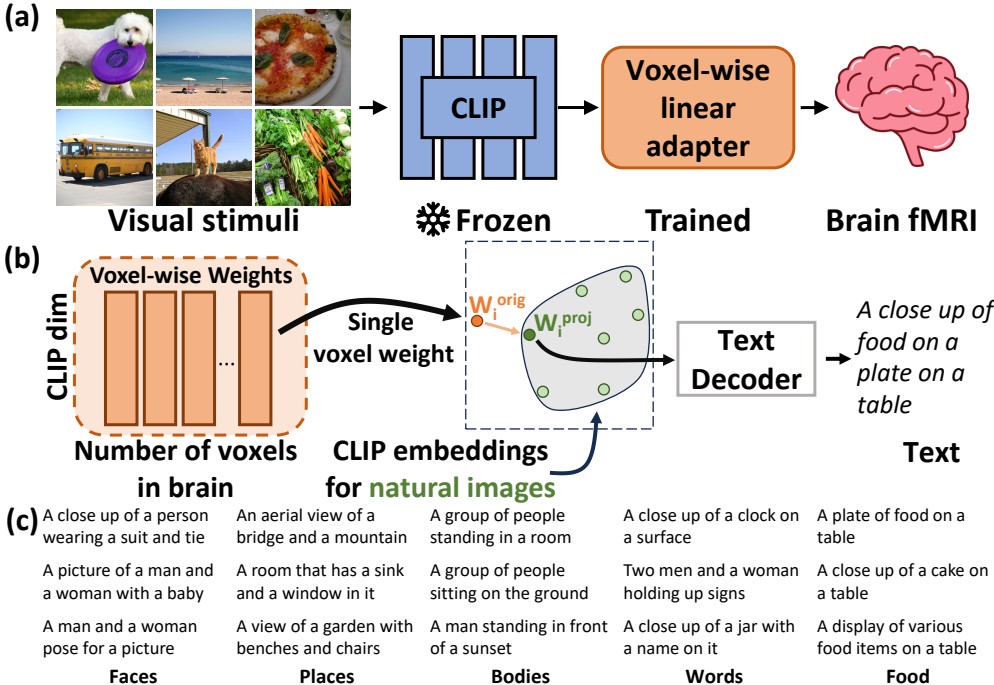

**Figure 1:** **Architecture of BrainSCUBA.** **(a)** Our framework relies on an fMRI encoder trained to map from images to voxel-wise brain activations. The encoder consists of a frozen CLIP image network with a unit norm output and a linear probe. **(b)** We decode the voxel-wise weights by projecting the weights into the space of CLIP embeddings for natural images followed by sentence generation. **(c)** Select sentences from each region, please see experiments for a full analysis.

Some approaches further utilize or generate captions that describe the observed visual stimuli (Doerig et al., 2022; Ferrante et al., 2023; Liu et al., 2023; Mai & Zhang, 2023).

The second approach seeks to generate stimuli that *activates* a given region rather than exactly reconstructing the input (Walker et al., 2019; Bashivan et al., 2019). Some of these approaches utilize GANs or Diffusion models to constrain the synthesized output (Ponce et al., 2019; Ratan Murty et al., 2021; Gu et al., 2022; Luo et al., 2023). BrainSCUBA falls under the broad umbrella of this second approach. But unlike prior methods which were restricted to modeling broad swathes of the brain, our method can be applied at voxel-level, and can output concrete interpretable captions.

## 3 METHODS

We aim to generate fine-grained (voxel-level) natural language captions that describe a visual scene which maximally activate a given voxel. We first describe the parameterization and training of our voxel-wise fMRI encoder which goes from images to brain activations. We then describe how we can analytically derive the optimal CLIP embedding given the encoder weights. Finally, we describe how we close the gap between optimal CLIP embeddings and the natural image embedding space to enable voxel-conditioned caption generation. We illustrate our framework in Figure 1.

### 3.1 IMAGE-TO-BRAIN ENCODER CONSTRUCTION

An image-computable brain encoder is a learned function $F_\theta$ that transforms an image $\mathcal{I} \in \mathbb{R}^{H \times W \times 3}$ to voxel-wise brain activation beta values represented as a $1D$ vector of $N$ brain voxels $B \in \mathbb{R}^{1 \times N}$, where $F_\theta(\mathcal{I}) \Rightarrow B$. Recent work identified models trained with a contrastive vision-language objective as the highest performing feature extractor for visual cortex, with later CLIP layers being more accurate for higher visual areas (Wang et al., 2022; Conwell et al., 2023). As we seek to solely model higher-order visual areas, we utilize a two part design for our encoder. First is a frozen CLIP (Radford et al., 2021) backbone which outputs a $R^{1 \times M}$ dimensional embedding vector for each image. The second is a linear probe $W \in \mathcal{R}^{M \times N}$ with bias $b \in \mathcal{R}^{1 \times N}$, which transform a unit-norm

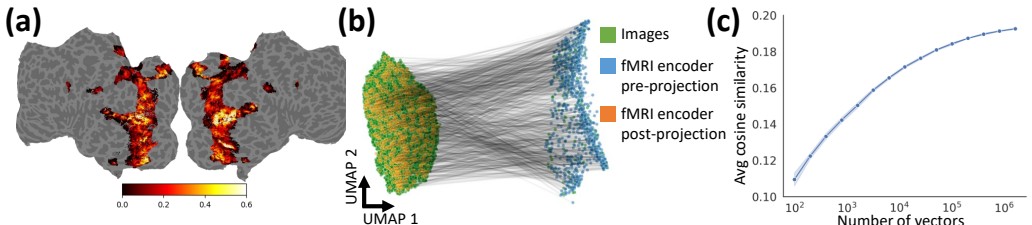

Figure 2: **Projection of fMRI encoder weights**. **(a)** We validate the encoder $R^2$ on a test set, and find it can achieve high accuracy in the higher visual cortex. **(b)** The joint-UMAP of image CLIP embeddings, and pre-/post-projection of the encoder. All embeddings are normalized before UMAP. **(c)** We measure the average cosine similarity between pre-/post-projection weights, and find it increases as the images used are increased. Standard deviation of 5 projections shown in light blue.

image embedding to brain activations.

$$\left[ \frac{\text{CLIP}_{\text{img}}(\mathcal{I})}{\|\text{CLIP}_{\text{img}}(\mathcal{I})\|_2} \times W + b \right] \Rightarrow B \tag{1}$$

After training with MSE loss, we evaluate the encoder on the test set in Figure 2(a) and find that our encoder can achieve high $R^2$.

### 3.2 DERIVING THE OPTIMAL EMBEDDING AND CLOSING THE GAP

The fMRI encoder we construct utilizes a linear probe applied to a unit-norm CLIP embedding. It follows from the design that the maximizing embedding $e_i^*$ for a voxel $i$ can be derived efficiently from the weight, and the predicted activation is upper bounded by $\|W_i\|_2 + b$ when

$$e_i^* = \frac{W_i}{\|W_i\|_2} \tag{2}$$

In practice, a natural image $\mathcal{I}^*$ that achieves $\frac{\text{CLIP}_{\text{img}}(\mathcal{I}^*)}{\|\text{CLIP}_{\text{img}}(\mathcal{I}^*)\|_2} = e_i^*$ does not typically exist. There is a modality gap between the CLIP embeddings of natural images and the optimal embedding derived from the linear weight matrix. We visualize this gap in Figure 2(b) in a joint UMAP (McInnes et al., 2018) fitted on CLIP ViT-B/32 embeddings and fMRI encoder weights, both normalized to unit-norm. To close this modality gap, we utilize a softmax weighted sum to project the voxel weights onto the space of natural images. Let the original voxel weight be $W_i^{\text{orig}} \in R^{1 \times M}$, which we will assume to be unit-norm for convenience. We have a set with $K$ natural images $M = \{M_1, M_2, M_3, \cdots, M_K\}$. For each image, we compute the CLIP embedding $e_j = \text{CLIP}_{\text{img}}(M_j)$. Given $W_i^{\text{orig}}$, we use cosine similarity followed by softmax with temperature $\tau$ to compute a score that sums to 1 across all images. For each weight $W_i^{\text{orig}}$ and example image $M_j$:

$$\text{Score}_{i,j} = \frac{\exp(W_i^{\text{orig}} e_j^T / \tau)}{\exp(\sum_{k=1}^{K} W_i^{\text{orig}} e_k^T / \tau)} \tag{3}$$

We parameterize $W_i^{\text{proj}}$ using a weighted sum derived from the scores, applied to the norms and directions of the image embeddings:

$$W_i^{\text{proj}} = \left( \sum_{k=1}^{K} \text{Score}_{i,k} * \|e_k\|_2 \right) * \left( \sum_{k=1}^{K} \text{Score}_{i,k} * \frac{e_k}{\|e_k\|_2} \right) \tag{4}$$

In Figure 2(c) we show the cosine similarity between $W_i^{\text{orig}}$ and $W_i^{\text{proj}}$ as we increase the size of $M$. This projection operator can be treated as a special case of dot-product attention (Vaswani et al., 2017), with query $= W_i^{\text{orig}}$, key $= \{e_1, e_2, \cdots, e_K\}$, and value equal to norm or direction of $\{e_1, e_2, \cdots, e_K\}$. A similar approach is leveraged by Li et al. (2023b), which shows a similar operator outperforms nearest neighbor search for text-only caption inference. As $W_i^{\text{proj}}$ lies in the space of CLIP embeddings for natural images, this allows us to leverage any existing captioning system that is solely conditioned on the final CLIP embedding of an image. We utilize a frozen CLIPCap network, consisting of a projection layer and finetuned GPT-2 (Mokady et al., 2021).

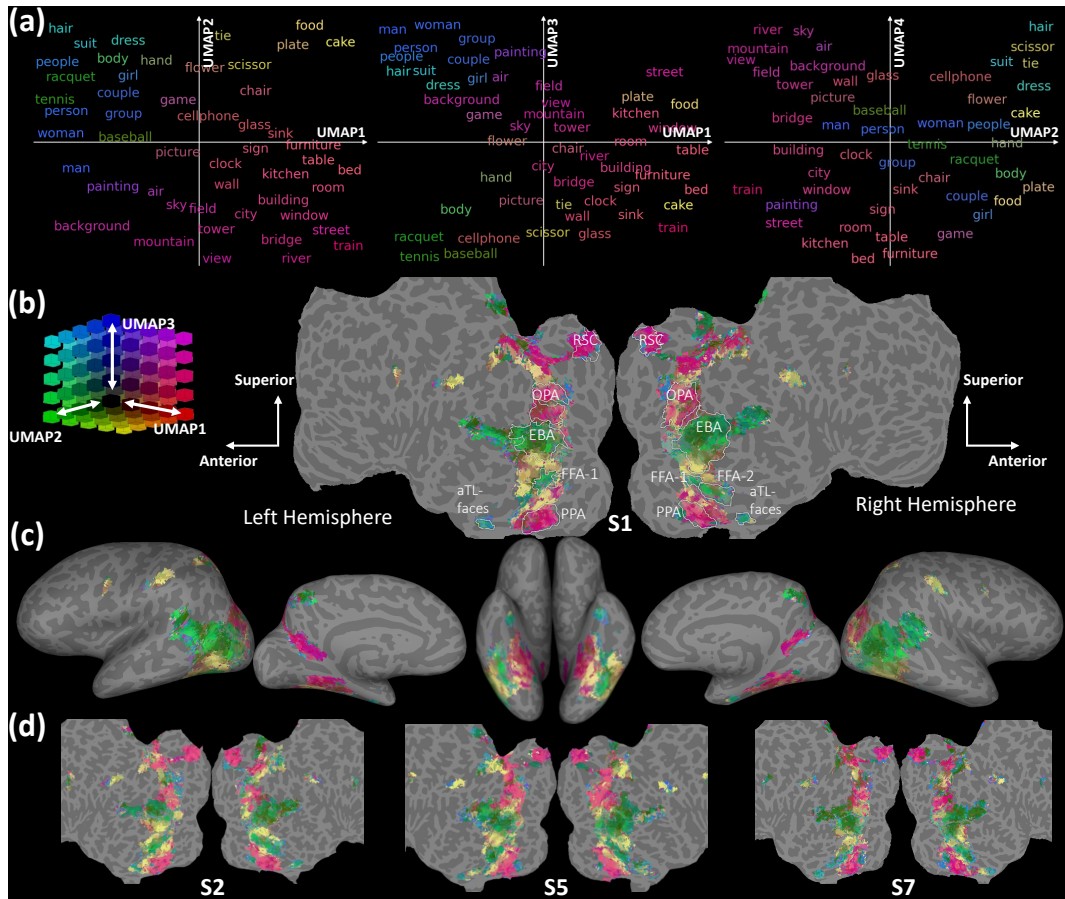

Figure 3: **Interpreting the nouns generated by BrainSCUBA** . We take the projected encoder weights and fit a UMAP transform that goes to 4-dims. **(a)** The 50 most common noun embeddings across the brain are projected & transformed using the fMRI UMAP. **(b)** Flatmap of **S1** with ROIs labeled. **(c)** Inflated view of **S1**. **(d)** Flatmaps of **S2, S5, S7**. We find that BrainSCUBA nouns are aligned to previously identified functional regions. Shown here are body regions (EBA), face regions (FFA-1/FFA-2/aTL-faces), place regions (RSC/OPA/PPA). Note that the yellow near FFA match the food regions identified by Jain et al. (2023). The visualization style is inspired by Huth et al. (2016).

## 4    RESULTS

In this section, we utilize BrainSCUBA to generate voxel-wise captions and demonstrate that it can capture the selectivity in different semantic regions in the brain. We first show that the generated nouns are interpretable across the entire brain and exhibit a high degree of specificity within pre-identified category-selective regions. Subsequently, we use the captions as input to text-to-image diffusion models to generate novel images, and confirm the images are semantically consistent within their respective regions. Finally, we utilize BrainSCUBA to analyze the distribution of person representations across the brain to offer novel neuroscientific insight. These results illustrate BrainSCUBA's ability to characterize human visual cortical populations, rendering it a promising framework for exploratory neuroscience.

### 4.1    SETUP

We utilize the Natural Scenes Dataset (NSD; Allen et al. (2022)), the largest whole-brain 7T human visual stimuli dataset. Of the 8 subjects, 4 subjects viewed the full $10,000$ image set repeated $3\times$. We use these subjects, S1, S2, S5, S7, for experiments in the main paper, and present additional results in the appendix. The fMRI activations (betas) are computed using GLMSingle (Prince et al., 2022), and further normalized so each voxel's response is $\mu = 0, \sigma^2 = 1$ on a session basis. The response across repeated viewings of the same image is averaged. The brain encoder is trained on the $\sim 9000$ unique images for each subject, while the remaining $\sim 1000$ images viewed by all are used to validate $R^2$.

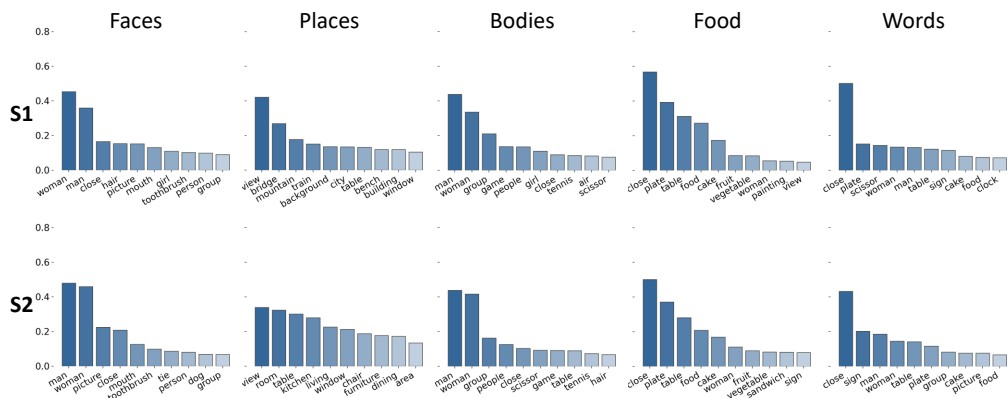

Figure 4: **Top BrainSCUBA nouns via voxel-wise captioning in broad category selective regions.** We perform part-of-speech tagging and lemmatization to extract the nouns, $y-$axis normalized by voxel count. We find that the generated captions are semantically related to the functional selectivity of broad category selective regions. Note that the word "close" tended to appear in the noun phrase "close-up", which explains its high frequency in the captions from food- and word-selective voxels.

The unpaired image projection set is a 2 million combination of LAION-A v2 (6+ subset) and Open Images (Schuhmann et al., 2022; Kuznetsova et al., 2020). We utilize OpenAI's ViT-B/32 for the encoder backbone and embedding computation as this is the standard for CLIP conditioned caption generation. For image generation, we use the same model as used by Luo et al. (2023) in BrainDiVE, stable-diffusion-2-1-base with 50 steps of second order DPM-Solver++. In order to ensure direct comparability with BrainDiVE results, OpenCLIP's CoCa ViT-L/14 is used for image retrieval and zero-shot classification. We define face/place/body/word regions using independent category localizer data provided with the NSD by Allen et al. (2022) (threshold of $t > 2$), and use the masks provided by Jain et al. (2023) to define the food regions. For details on the human study, please see the appendix.

## 4.2 VOXEL-WISE TEXT GENERATIONS

In this section, we first investigate how BrainSCUBA outputs conceptually tile the higher visual cortex. We perform part-of-speech (POS) tagging and lemmatization of the BrainSCUBA output for four subjects, and extract the top-50 nouns. To extract noun specific CLIP embeddings, we reconstitute them into sentences of the form "A photo of a/an [NOUN]" as suggested by CLIP. Both the noun embeddings and the brain encoder voxel-wise weights are projected to the space of CLIP image embeddings and normalized to the unit-sphere for UMAP. We utilize UMAP fit on the encoder weights for S1. Results are shown in Figure 3. We observe that the nouns generated by BrainSCUBA are conceptually aligned to pre-identified functional regions. Namely, voxels in extrastriate body area (EBA) are selective to nouns that indicate bodies and activities (green), fusiform face area (FFA-1/FFA-2) exhibits person/body noun selectivity (blue-green), place regions – retrosplenial cortex (RSC), occipital place area (OPA), and parahippocampal place area (PPA) – show selectivity for scene elements (magenta), and the food regions (yellow; Jain et al. (2023)) surrounding FFA exhibit selectivity for food-related nouns. These results show that our framework can characterize the broad semantic selectivity of visual cortex in a zero-shot fashion.

We further quantify the top-10 nouns within each broad category selective region (Figure 4). We observe that BrainSCUBA generates nouns that are conceptually matched to the expected preferred category of each region. Note the multimodal selectivity for words/people/food within the word region has also been observed by Mei et al. (2010); Khosla & Wehbe (2022).

## 4.3 TEXT-GUIDED BRAIN IMAGE SYNTHESIS

Visualization of the captions can be helpful in highlighting subtle co-occurrence statistics, with novel images critical for future hypothesis driven investigations of the visual cortex (Ratan Murty et al., 2021; Gu et al., 2023; Jain et al., 2023). We utilize a text-to-image diffusion model, and condition the synthesis process on the voxel-wise captions within an ROI (Figure 5). We perform 1000 generations per-ROI, subsampling without replacement when the number of voxels/captions in a ROI exceed 1000, and randomly sample the gap when there are fewer than 1000. For face-, place-, word-, body-selective regions, we visualize the top-5 out of 10, 000 images ranked by real

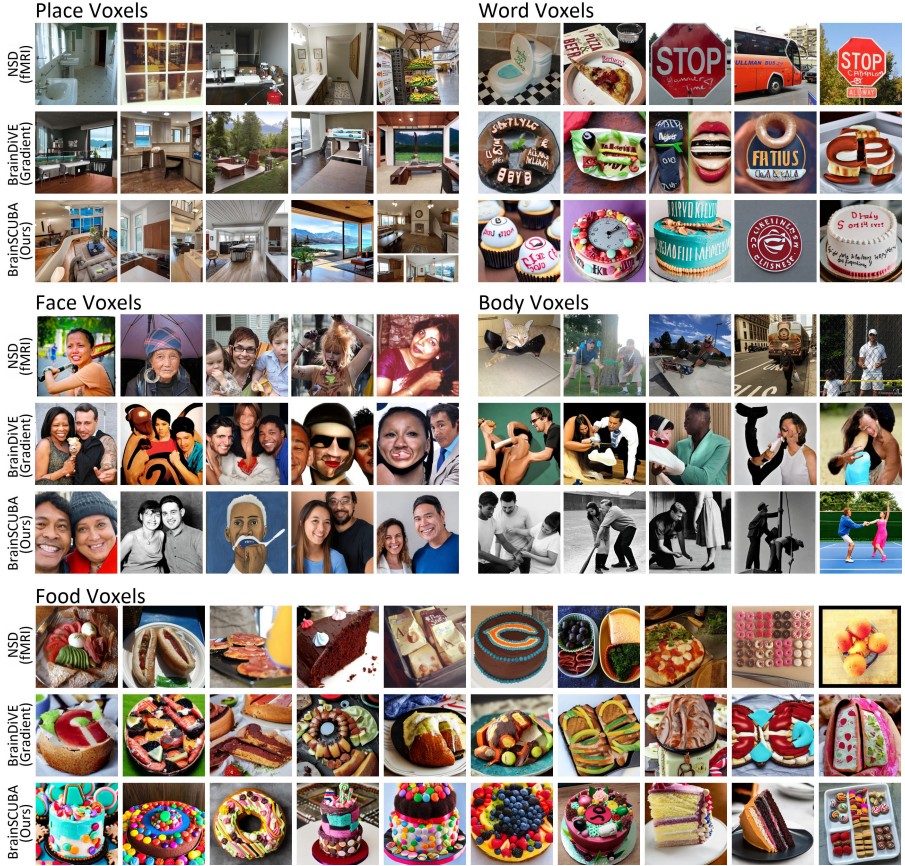

Figure 5: **Novel images for category selective voxels in S2**. We visualize the top-5 images from the fMRI stimuli and generated images for the place/word/face/body regions, and the top-10 images for the food region. We observe that images generated with BrainSCUBA appear more coherent.

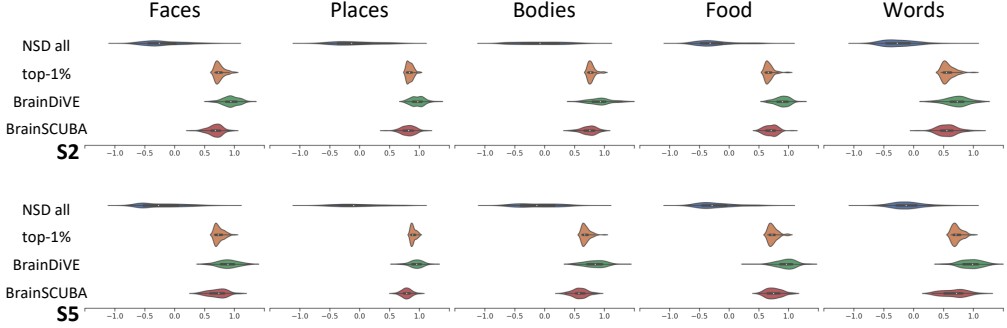

Figure 6: **Evaluating the distribution of BrainSCUBA captions with a different encoder**. We train an encoder with a different backbone (EVA02-CLIP-B-16) from both BrainDiVE and BrainSCUBA. For each region, we evaluate the response to all images a subject saw in NSD, the response of the top-1% of images in NSD stimuli ranked using `EVA02`, the top-10% of images generated by BrainDiVE and BrainSCUBA and ranked by their respective encoders. Each region is normalized to $[-1, 1]$ using the min/max of the predicted responses to NSD stimuli. BrainSCUBA can achieve high predicted responses despite *not* performing explicit gradient based maximization like BrainDiVE, and yields concretely interpretable captions. BrainSCUBA is also $\sim 10\times$ faster per image.

average ROI response from the fMRI stimuli (NSD), and the top-5 out of $1,000$ generations ranked by predicted response using the respective BrainDiVE (Luo et al., 2023) and BrainSCUBA encoders. BrainDiVE is used for comparison as it is the state of the art method for synthesizing activating images in the higher visual cortex, and we follow their evaluation procedure. For the food region, we visualize the top-10. Predicted activation is shown in Figure 6, with semantic classification shown in Table 1. Visual inspection suggests our method can generate diverse images semantically aligned

|  | Faces | | Places | | Bodies | | Words | | Food | | Mean | |
|---|---|---|---|---|---|---|---|---|---|---|---|---|
|  | S2 | S5 | S2 | S5 | S2 | S5 | S2 | S5 | S2 | S5 | S2 | S5 |
| NSD all stim | 17.1 | 17.5 | 29.4 | 30.7 | 31.5 | 30.3 | 11.0 | 10.1 | 10.9 | 11.4 | 20.0 | 20.0 |
| NSD top-100 | 45.0 | 43.0 | 78.0 | 93.0 | 59.0 | 55.0 | 48.0 | 33.0 | 86.0 | 83.0 | 63.2 | 61.4 |
| BrainDiVE-100 | 68.0 | 64.0 | 100 | 100 | 69.0 | 77.0 | 61.0 | 80.0 | 94.0 | 87.0 | 78.4 | 81.6 |
| **BrainSCUBA-100** | **67.0** | **62.0** | **100** | **99.0** | **54.0** | **73.0** | **55.0** | **34.0** | **97.0** | **92.0** | **74.6** | **72.0** |

Table 1: **Semantic evaluation of images with zero-shot CLIP.** We use CLIP to perform zero-shot 5-way classification. Show here is the percentage where category of the image matches the preferred category for a brain region. This is shown for each subject's NSD stimulus set ($10,000$ images for S2&S5); the top-100 images (top-1%) evaluated by average region true fMRI, the top-100 (10%) of BrainDiVE and BrainSCUBA (**bolded**) as evaluated by their respective encoders. BrainSCUBA has selectivity that is closer to the true NSD top 1%.

with the target category. Our images are generally more visually coherent than those generated by BrainDiVE, and contain more clear text in word voxels, and fewer degraded faces and bodies in the respective regions. This is likely because our images are conditioned on text, while BrainDiVE utilizes the gradient signal alone.

## 4.4 INVESTIGATING THE BRAIN'S SOCIAL NETWORK

The intrinsic social nature of humans significantly influences visual perception. This interplay is evident in the heightened visual sensitivity towards social entities such as faces and bodies (Pitcher & Ungerleider, 2021; Kanwisher et al., 1997; Downing et al., 2001). In this section, we explore if BrainSCUBA can provide insights on the finer-grained coding of people in the brain. We use a rule based filter and count the number of captions that contain one of 140 nouns that describe people (person, man, woman, child, boy, girl, family, occupations, and plurals). We visualize the voxels whose captions contain people in Figure 7, and provide a quantitative evaluation in Table 2. We observe that our captions can correctly identify non-person-selective scene, food, and word regions as having lower person content than person-selective ROIs like the FFA or the EBA. Going beyond traditional functional ROIs, we find that the precuneus visual area (PCV) and the temporoparietal junction (TPJ) have a very high density of captions with people. The precuneus has been implicated in third-person mental representations of self (Cavanna & Trimble, 2006; Petrini et al., 2014), while the TPJ has been suggested to be involved in theory of mind and social cognition (Saxe & Kanwisher, 2013). Our results lend support to these hypotheses.

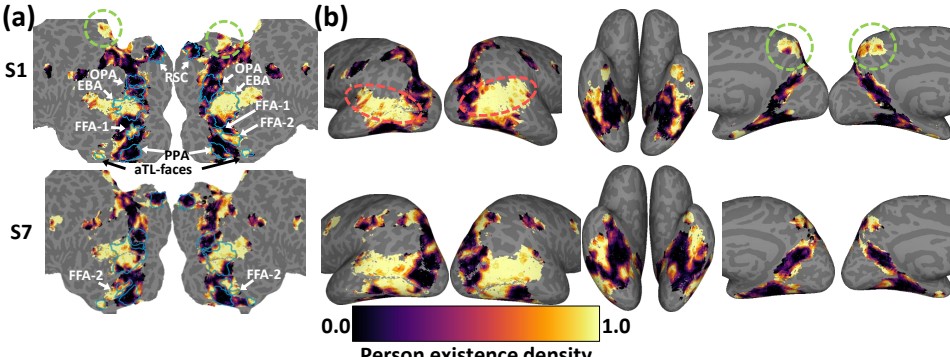

Figure 7: **Presence of people in captions**. We perform rule-based filtering and identify voxels where a caption contains at least one person. Data is surface smoothed for visualization. Dotted orange oval shows approximate location of TPJ, which is linked to theory of mind; green circle shows location of PCV, associated with third-person perspective of social interactions. Note that TPJ is HCP defined following Igelström & Graziano (2017), while PCV is HCP atlas region 27 (Glasser et al., 2016).

A close visual examination of Figure 7 suggests a divide within EBA. We perform spherical k-means clustering on joint encoder weights for $t > 2$ EBA from S1/S2/S5/S7, and identify two stable clusters. These clusters are visualized in Figure 8. Utilizing the rule parser, we labels the voxels into those that contain a single individual or multiple people, and further visualize the top-nouns within each of these two clusters. While both clusters include general person words like "man" and "woman", cluster 1 has more nouns that suggest groups of people interacting together (group, game, people), and cluster 2 has words that suggest close-ups of individuals with objects that may be hand-held. To validate our findings, we perform a study where subjects are asked to evaluate the top-100 images from each of the clusters. Results are shown in Table 3. Aligned with the top-nouns, the study suggests that

|  | Non-Person | | | | | Person | | Other | |
|---|---|---|---|---|---|---|---|---|---|
|  | RSC | OPA | PPA | Food | Word | EBA | FFA | PCV | TPJ |
| S1 | 12.9 | 17.3 | 10.6 | 11.5 | 32.0 | 87.2 | 88.5 | 89.7 | 92.1 |
| S2 | 5.58 | 8.15 | 2.70 | 20.0 | 34.8 | 81.4 | 87.2 | 70.8 | 89.1 |
| S5 | 9.31 | 6.43 | 1.95 | 17.8 | 38.4 | 79.5 | 89.4 | 78.5 | 79.9 |
| S7 | 7.14 | 9.87 | 5.99 | 10.7 | 36.9 | 84.3 | 89.5 | 84.2 | 90.3 |
| Mean | 8.72 | 10.4 | 5.30 | 15.0 | 35.5 | 83.1 | 88.6 | 80.8 | 87.8 |

Table 2: **Percentage of captions in each region that contain people.** We observe a sharp difference between non-person regions (Scene RSC/OPA/PPA, Food, Word), and regions that are believed to be person selective (body EBA, face FFA). We also observe extremely high person density in PCV — a region involved in third-person social interactions, and TPJ — a region involved in social self-other distinction.

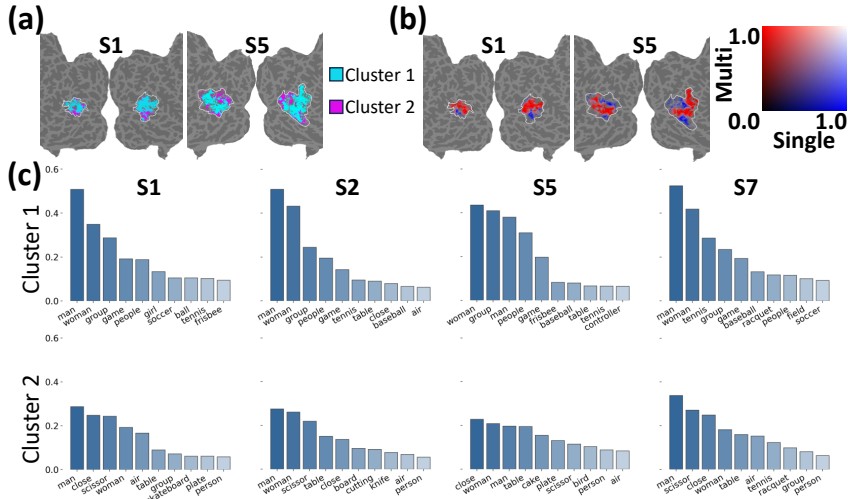

Figure 8: **Clusters within EBA. (a)** The EBA clusters for two subjects are shown on a flatmap. **(b)** Number of people mentioned in each caption. **(c)** Top nouns within each cluster, y-axis is normalized to the number of voxels within a cluster. Compared to Cluster-1, Cluster-2 has less emphasis on multiple people and more emphasis on objects that can be held.

| Which cluster is more... | people per-img | | | | inanimate objs | | | | far away | | | | sports | | | |
|---|---|---|---|---|---|---|---|---|---|---|---|---|---|---|---|---|
|  | S1 | S2 | S5 | S7 | S1 | S2 | S5 | S7 | S1 | S2 | S5 | S7 | S1 | S2 | S5 | S7 |
| EBA-1 (Cluster 1) | **88** | **84** | **91** | **78** | 15 | 11 | 12 | 13 | **62** | **72** | **78** | **63** | **75** | **79** | **85** | **76** |
| EBA-2 (Cluster 2) | 5 | 10 | 4 | 13 | **72** | **80** | **81** | **65** | 21 | 21 | 14 | 25 | 9 | 12 | 6 | 11 |

Table 3: **Human evaluation comparing two EBA clusters..** Evaluators compare the top 100 images for each cluster, with questions like "Which group of images is more X?", answers include EBA-1/EBA-2/Same. We do not show "Same"; responses sum to 100 across all three options. Results in %.

cluster-1 has more groups of people, fewer inanimate objects, and consists of larger scenes. This intriguing novel finding about the fine-grained distinctions in EBA can lead to new hypotheses about its function. This finding also demonstrates the ability of BrainSCUBA to uncover broad functional differences across the visual cortex.

## 5 DISCUSSION

**Limitations and Future Work.** Although our methods can generate semantically faithful descriptions for the broad category selective regions, our approach ultimately relies on a pre-trained captioning model. Due to this, our method reflects the biases of the captioning model. It is further not clear if the most selective object in each region can be perfectly captured by language. Future work could explore the use of more unconstrained captioning models (Tewel et al., 2022) or more powerful language models (Touvron et al., 2023).

**Conclusion.** To summarize, in this paper we propose BrainSCUBA, a method which can generate voxel-wise captions to describe each voxel's semantic selectivity. We explore how the output tiles the higher visual cortex, perform text-conditioned image synthesis with the captions, and apply it to uncover finer-grained patterns of selectivity in the brain within the person class. Our results suggest that BrainSCUBA may be used to facilitate data-driven exploration of the visual cortex.

## 6    ETHICS

Institutional review board approval was requested and granted for the human study on body region images. The NSD dataset contains 8 subjects from age 19 to 32; half white, half asian; 2 men, 6 women. These demographics are not necessarily reflective of the broader population. As part of our framework, we use a pretrained language model based on fine-tuned GPT-2, and will reflect the biases present in the training and fine-tuning data. We believe language models trained on more diverse datasets, or those trained with weaker constraints will help mitigate issues of bias. We also note that text-conditioned image diffusion models can reflect negative stereotypes, but such issues could be mitigated on a case-by-case basis using concept editing (Gandikota et al., 2024).

## 7    ACKNOWLEDGMENTS

This work used Bridges-2 at Pittsburgh Supercomputing Center through allocation SOC220017 from the Advanced Cyberinfrastructure Coordination Ecosystem: Services & Support (ACCESS) program, which is supported by National Science Foundation grants #2138259, #2138286, #2138307, #2137603, and #2138296. We also thank the Carnegie Mellon University Neuroscience Institute for support.

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

## A APPENDIX

## Sections

A.1   VISUALIZATION OF EACH SUBJECT'S TOP-NOUNS FOR CATEGORY SELECTIVE VOXELS

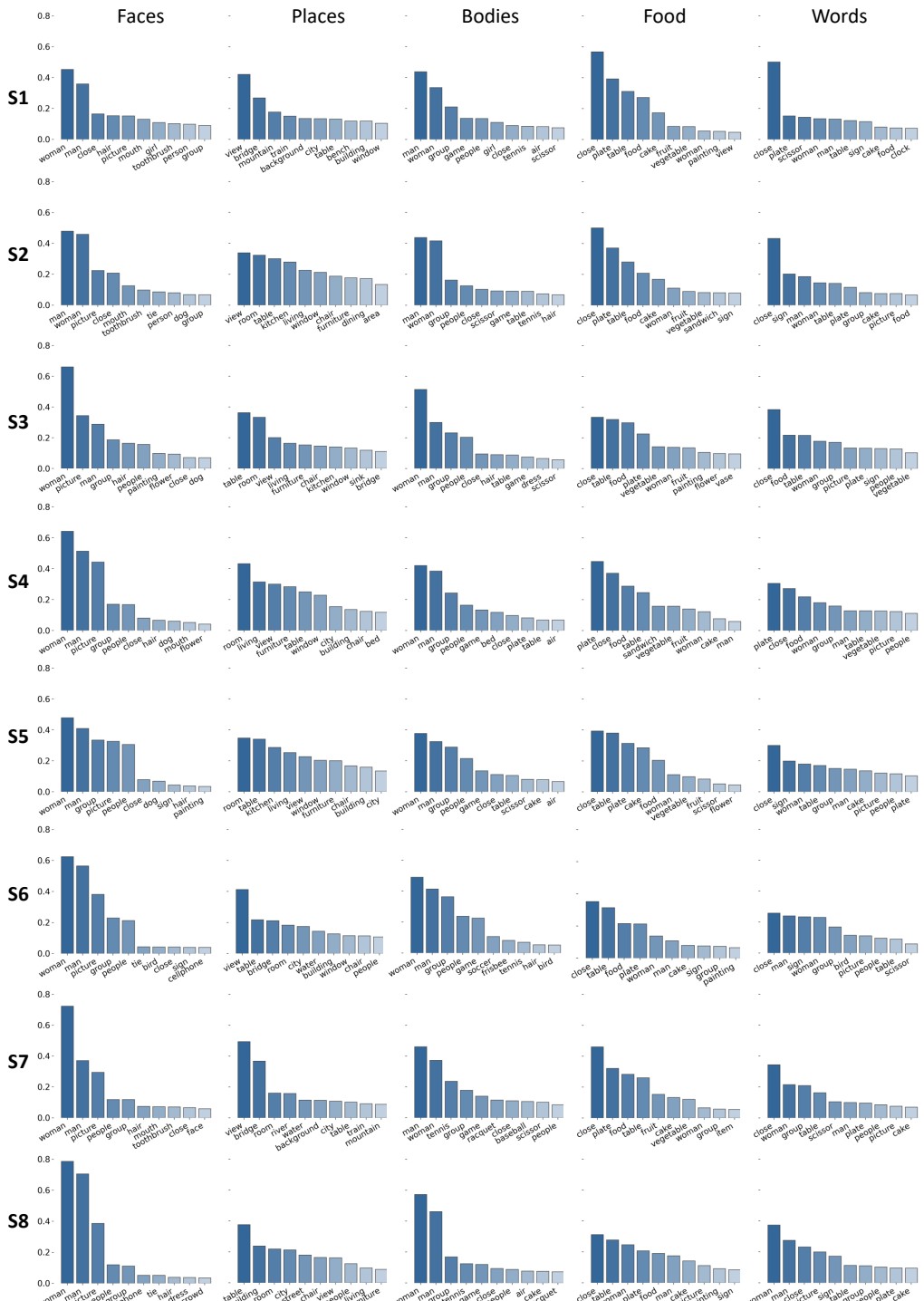

Figure S.1: **Top BrainSCUBA nouns via voxel-wise captioning in broad category selective regions for all subjects.** We see broad semantic alignment between the top-nouns and the semantic selectivity of a region. Note that the category selective voxels were derived from the intersection of official NSD functional localizer values $t > 2$ and their provided region masks.

## A.2 Visualization of UMAPs for all subjects

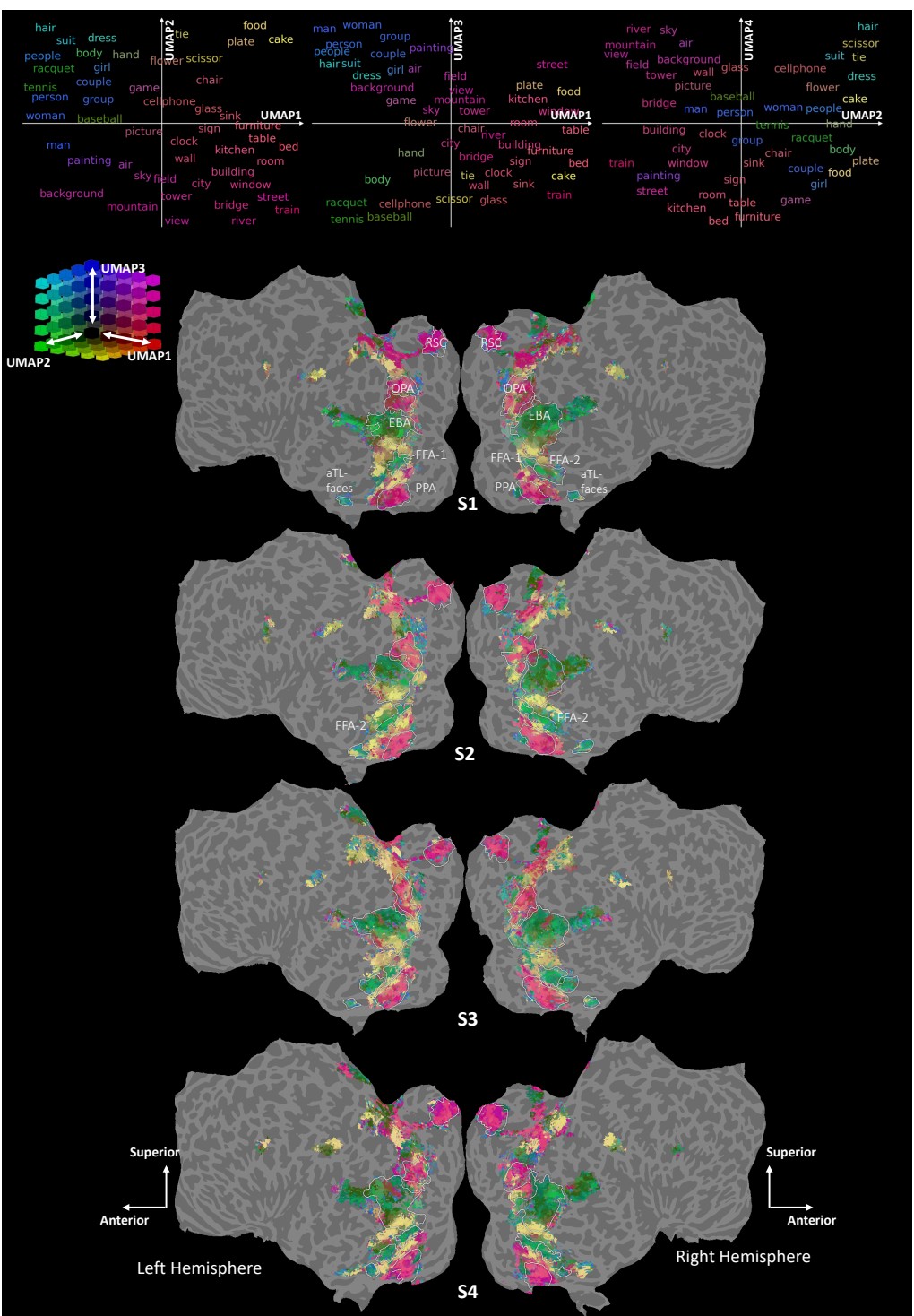

Figure S.2: **UMAP transform results for S1-S4.** All vectors are normalized to unit norm prior to UMAP. UMAP is fit on S1. Both word and voxel vectors are projected onto the space of natural images prior to transform using softmax weighted sum. Nouns are the most common across all subjects.

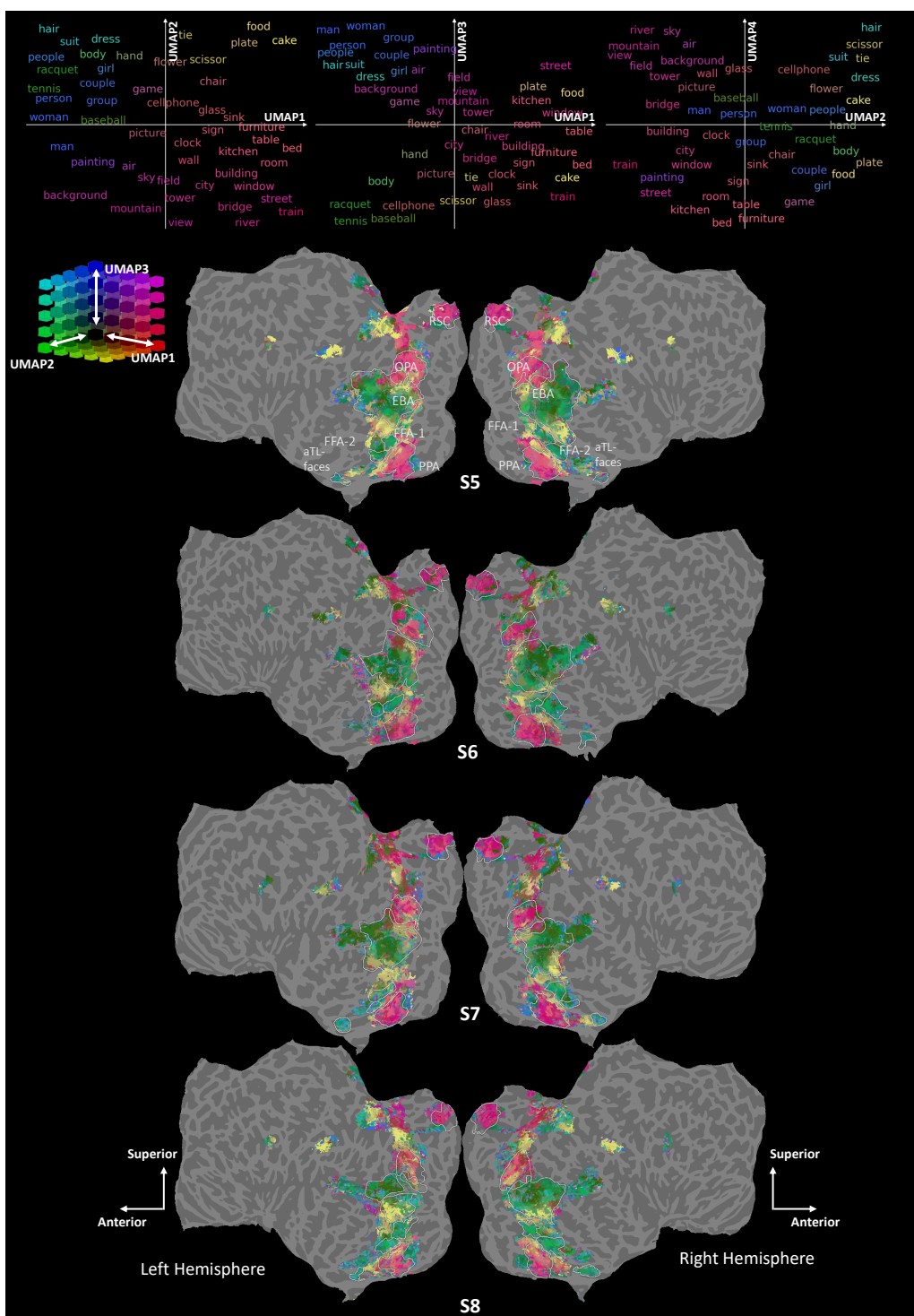

Figure S.3: **UMAP transform results for S5-S8.** All vectors are normalized to unit norm prior to UMAP. UMAP is fit on S1. Both word and voxel vectors are projected onto the space of natural images prior to transform using softmax weighted sum. Nouns are the most common across all subjects.

A.3    NOVEL IMAGE GENERATION FOR ALL SUBJECTS

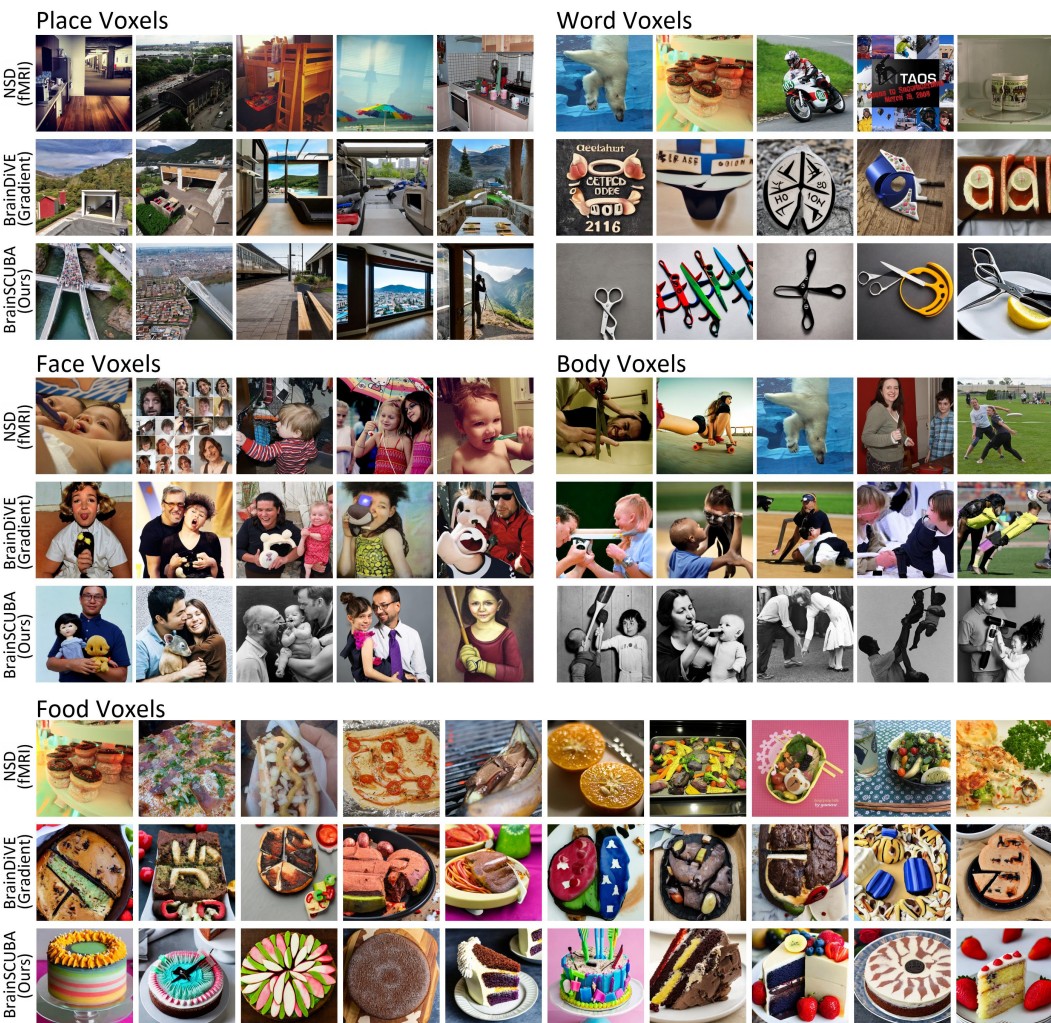

Figure S.4: **Image generation for S1.** We visualize the top-5 for face/place/body/word categories, and the top-10 for food. NSD images are ranked by ground truth response. BrainDiVE and BrainSCUBA are ranked by their respective encoders. BrainSCUBA images have more recognizable objects and fewer artifacts, likely due to the use of captions rather than gradients as in BrainDiVE.

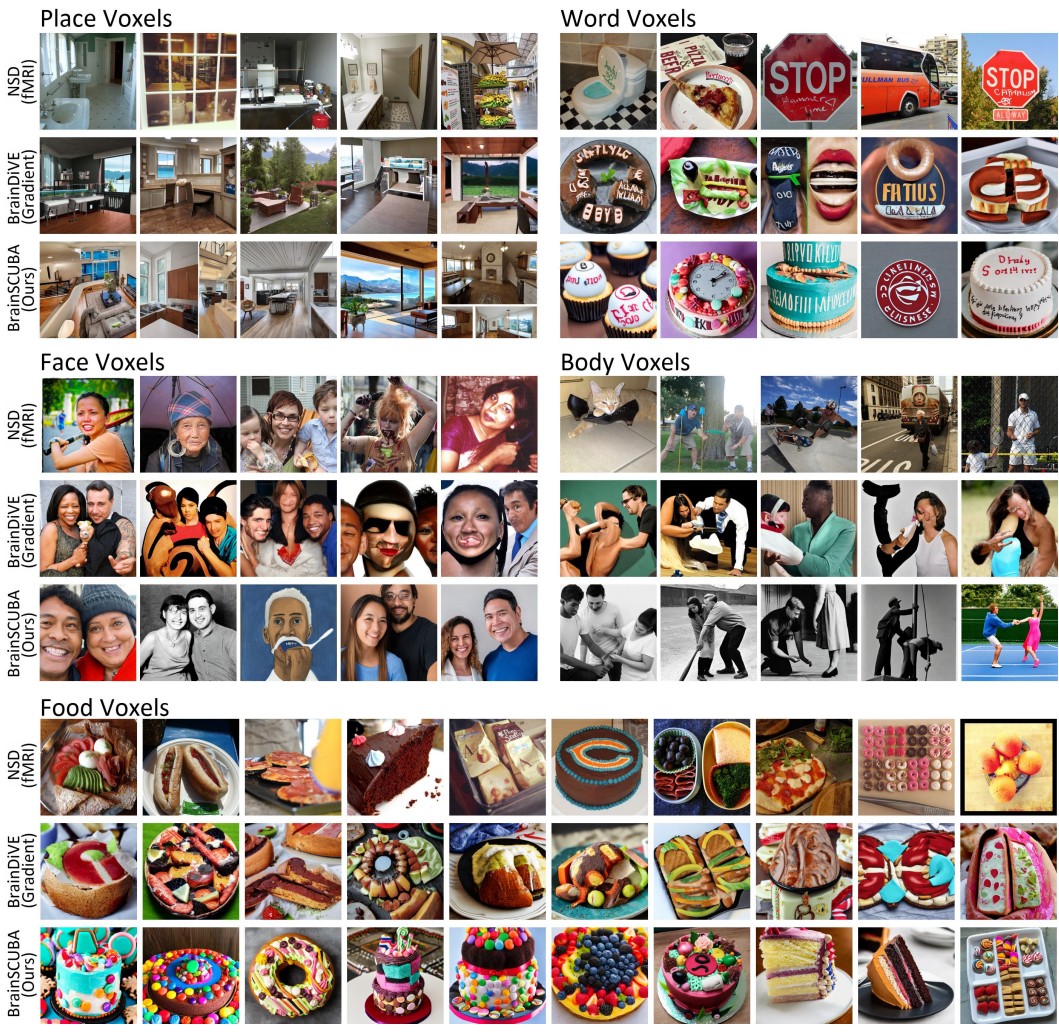

Figure S.5: **Image generation for S2.** We visualize the top-5 for face/place/body/word categories, and the top-10 for food. NSD images are ranked by ground truth response. BrainDiVE and BrainSCUBA are ranked by their respective encoders. BrainSCUBA images have more recognizable objects and fewer artifacts, likely due to the use of captions rather than gradients as in BrainDiVE.

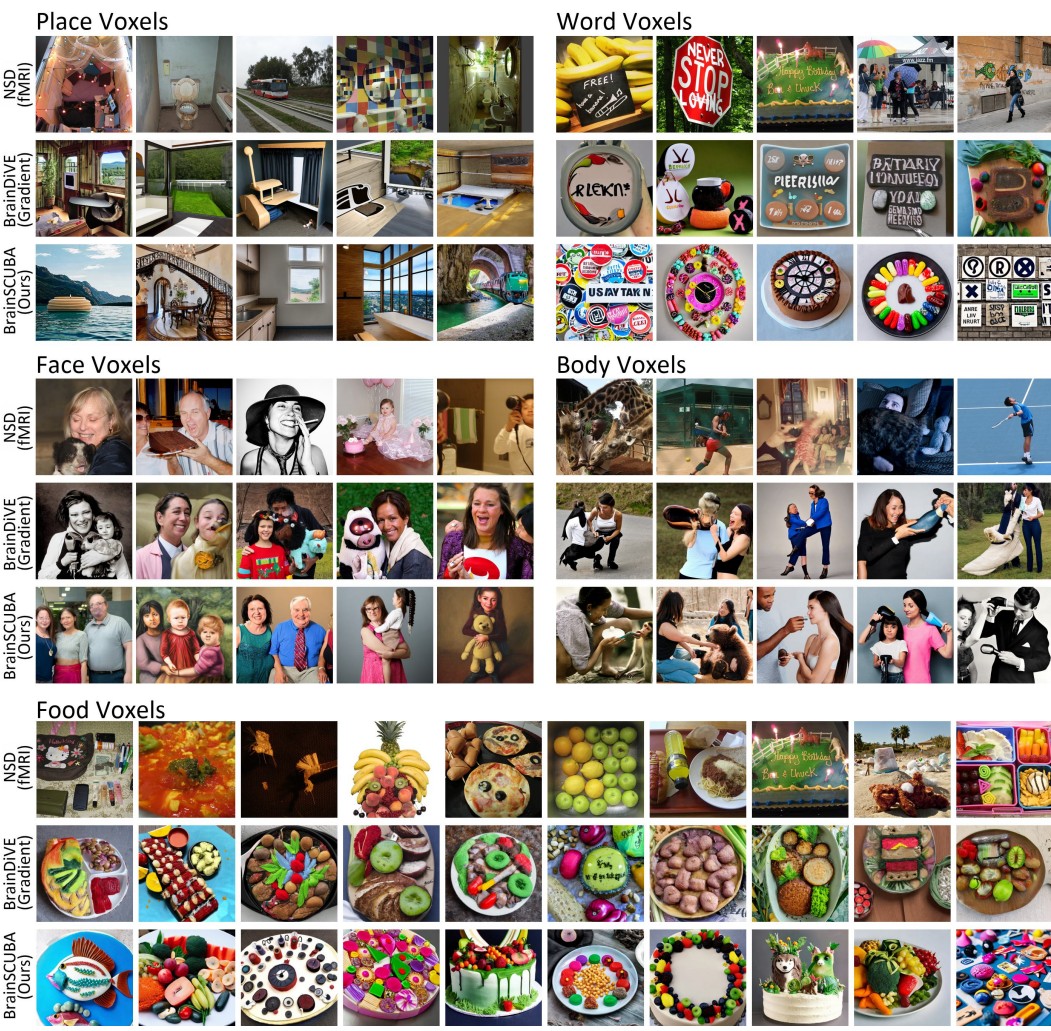

Figure S.6: **Image generation for S3.** We visualize the top-5 for face/place/body/word categories, and the top-10 for food. NSD images are ranked by ground truth response. BrainDiVE and BrainSCUBA are ranked by their respective encoders. BrainSCUBA images have more recognizable objects and fewer artifacts, likely due to the use of captions rather than gradients as in BrainDiVE.

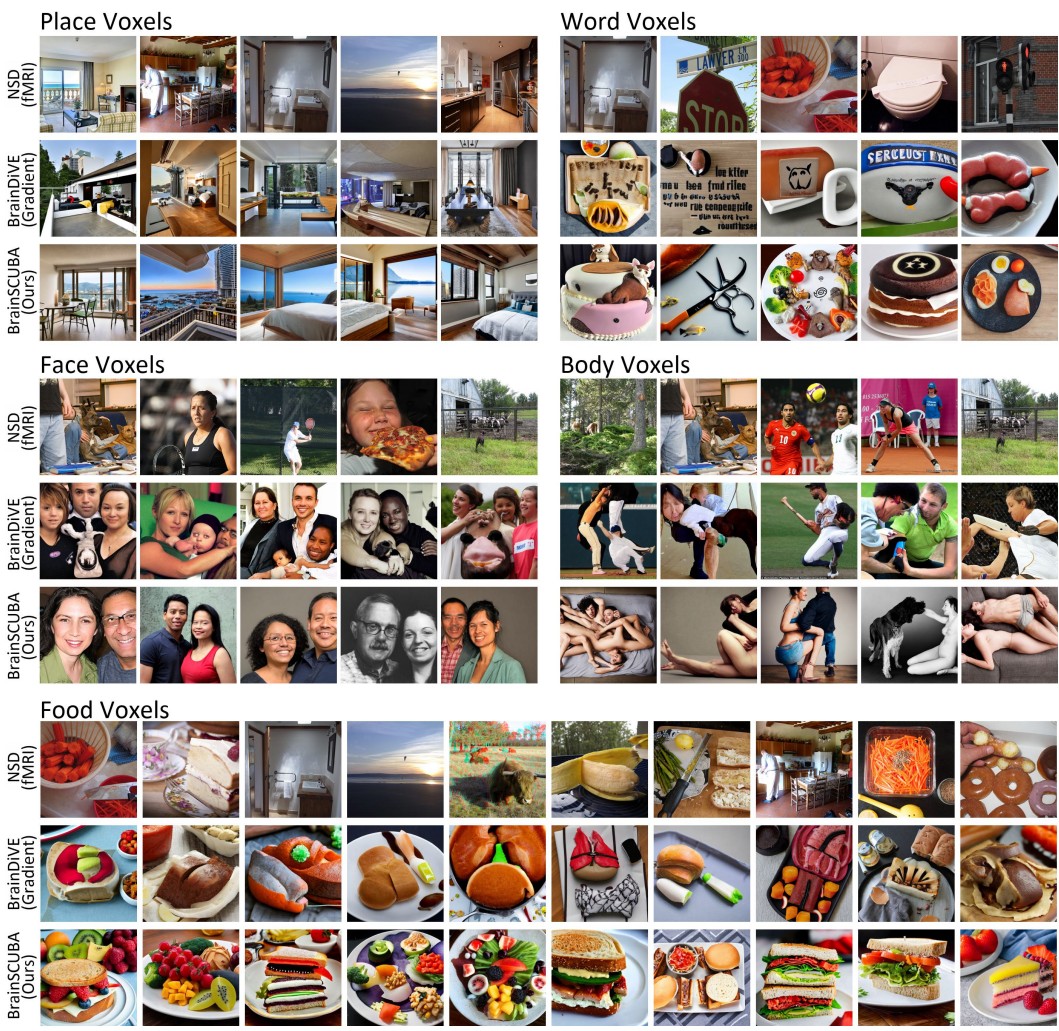

Figure S.7: **Image generation for S4.** We visualize the top-5 for face/place/body/word categories, and the top-10 for food. NSD images are ranked by ground truth response. BrainDiVE and BrainSCUBA are ranked by their respective encoders. BrainSCUBA images have more recognizable objects and fewer artifacts, likely due to the use of captions rather than gradients as in BrainDiVE.

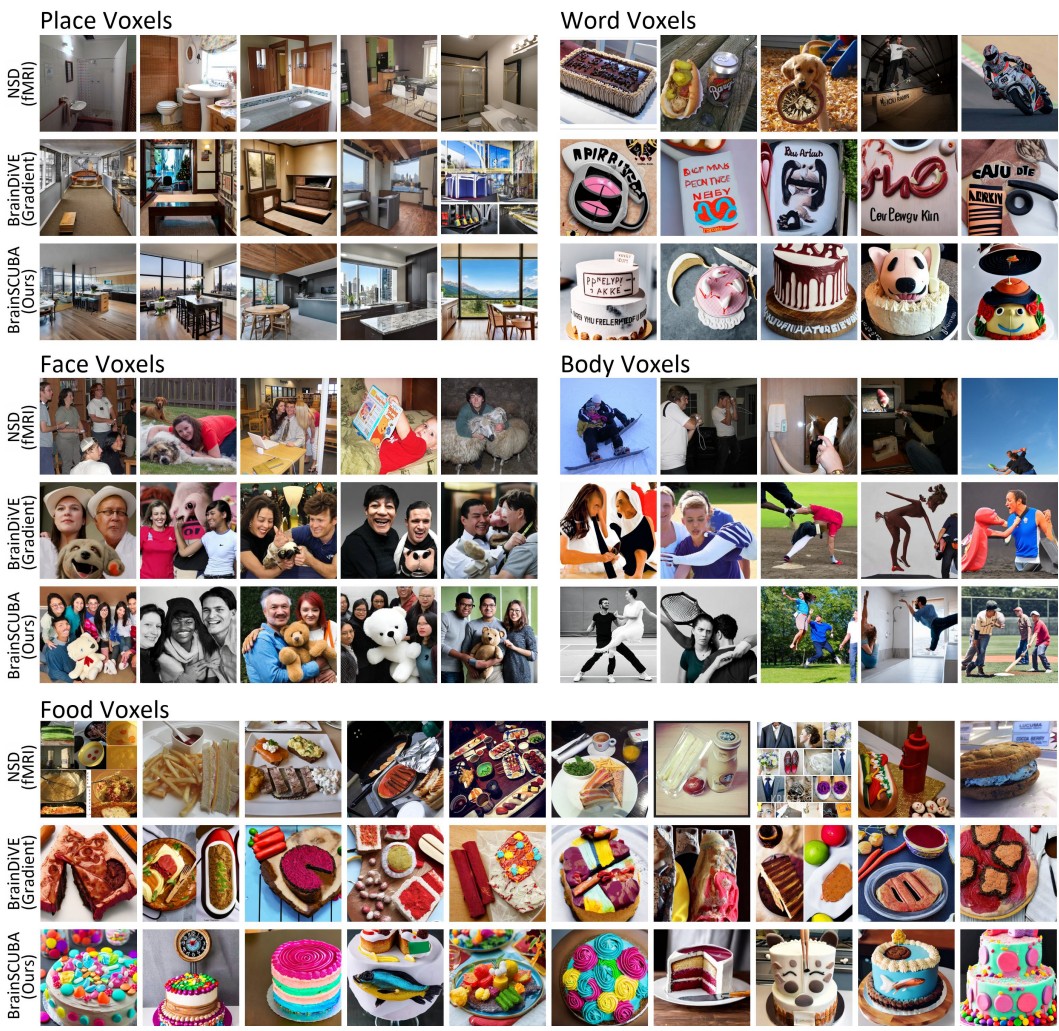

Figure S.8: **Image generation for S5.** We visualize the top-5 for face/place/body/word categories, and the top-10 for food. NSD images are ranked by ground truth response. BrainDiVE and BrainSCUBA are ranked by their respective encoders. BrainSCUBA images have more recognizable objects and fewer artifacts, likely due to the use of captions rather than gradients as in BrainDiVE.

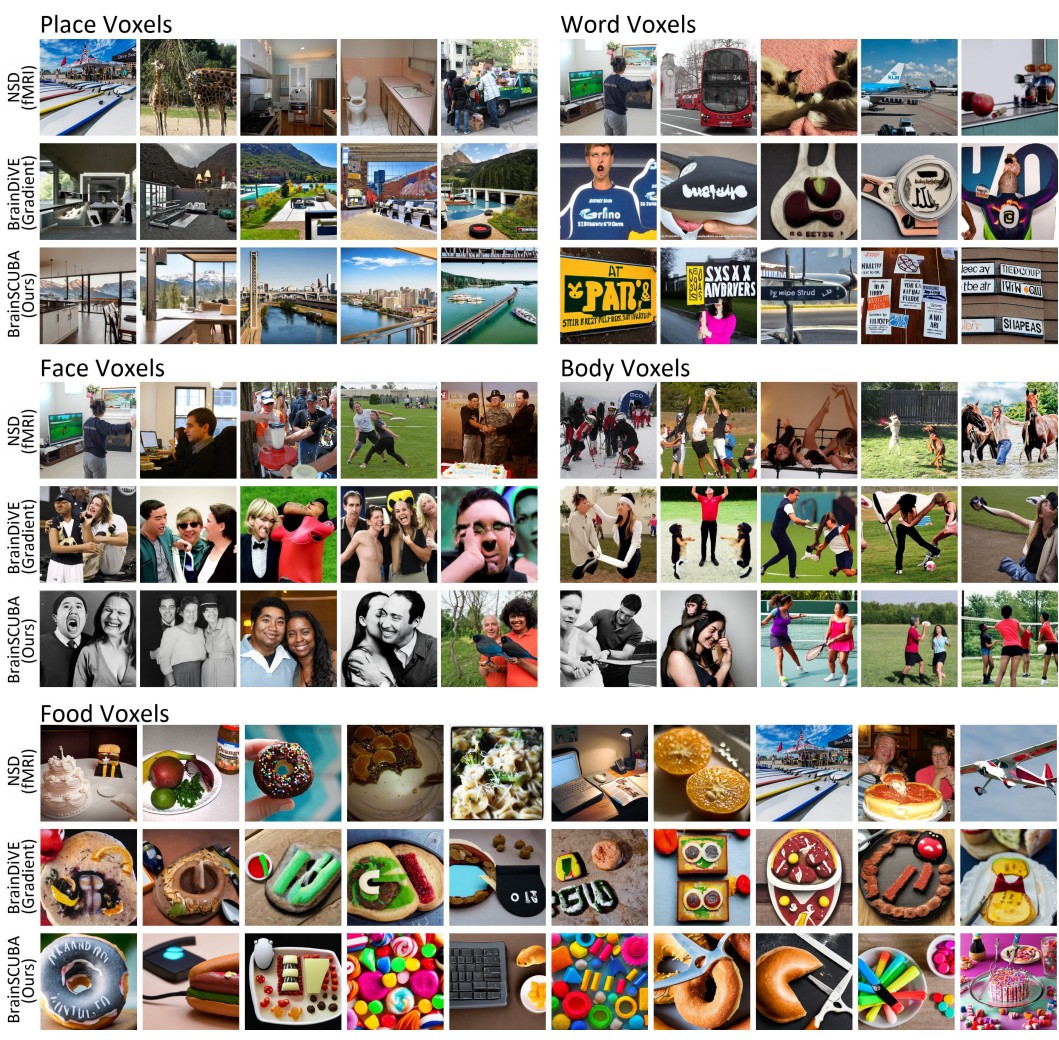

Figure S.9: **Image generation for S6.** We visualize the top-5 for face/place/body/word categories, and the top-10 for food. NSD images are ranked by ground truth response. BrainDiVE and BrainSCUBA are ranked by their respective encoders. BrainSCUBA images have more recognizable objects and fewer artifacts, likely due to the use of captions rather than gradients as in BrainDiVE.

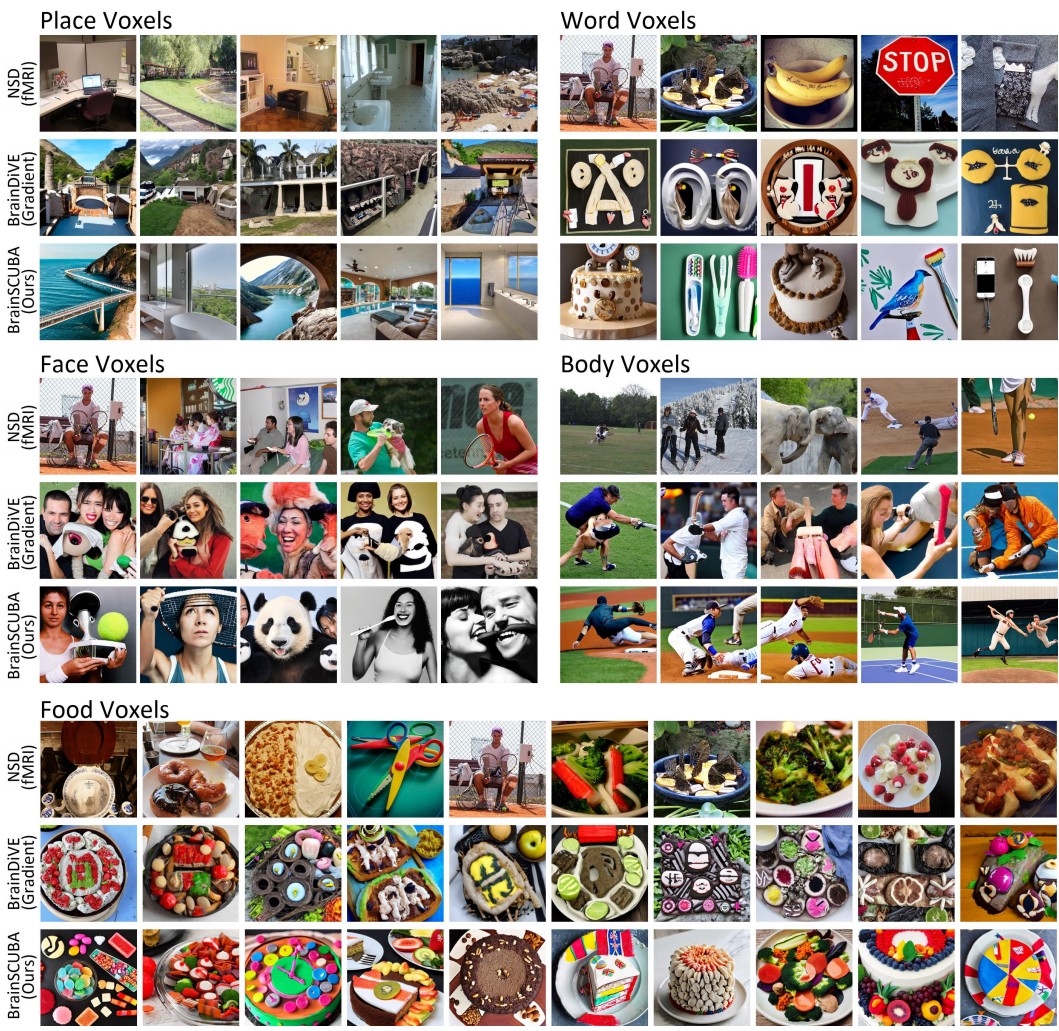

Figure S.10: **Image generation for S7.** We visualize the top-5 for face/place/body/word categories, and the top-10 for food. NSD images are ranked by ground truth response. BrainDiVE and Brain-SCUBA are ranked by their respective encoders. BrainSCUBA images have more recognizable objects and fewer artifacts, likely due to the use of captions rather than gradients as in BrainDiVE.

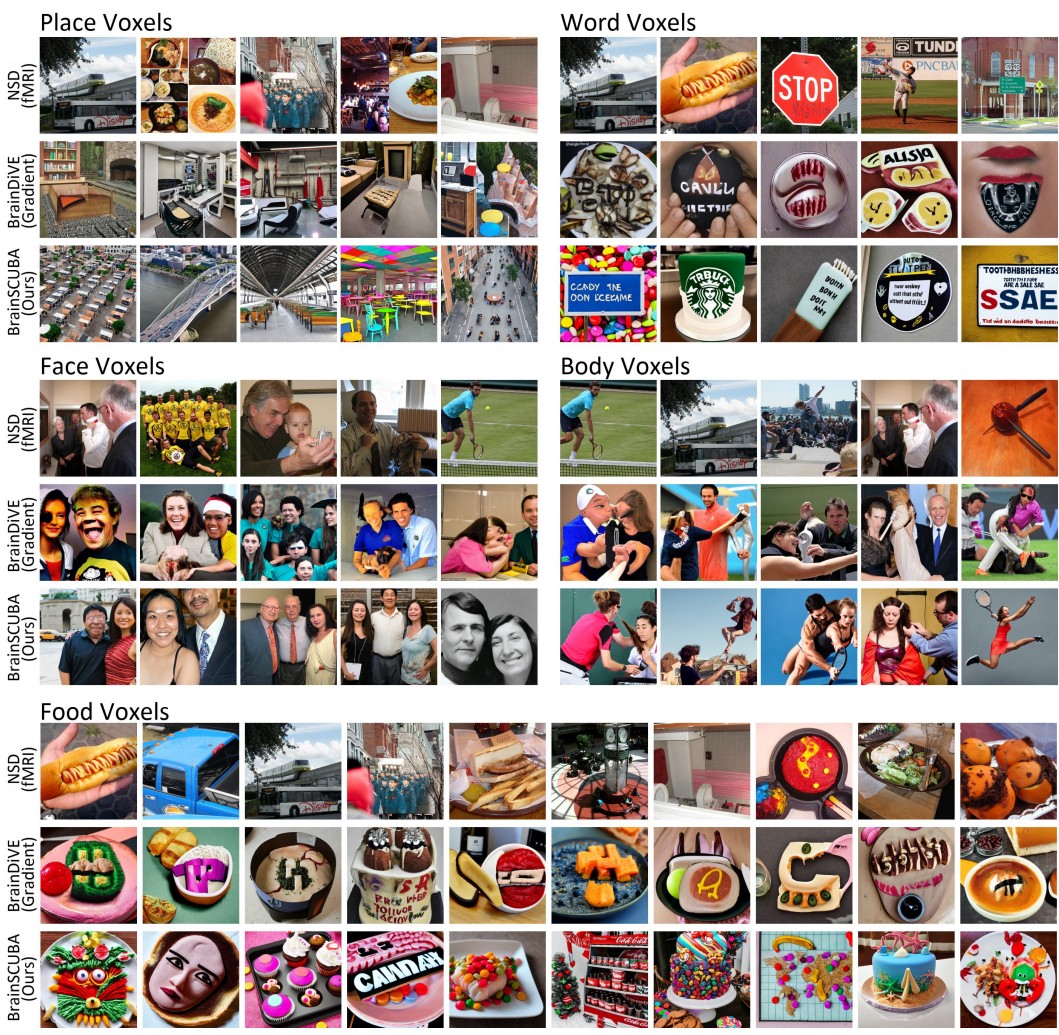

Figure S.11: **Image generation for S8.** We visualize the top-5 for face/place/body/word categories, and the top-10 for food. NSD images are ranked by ground truth response. BrainDiVE and Brain-SCUBA are ranked by their respective encoders. BrainSCUBA images have more recognizable objects and fewer artifacts, likely due to the use of captions rather than gradients as in BrainDiVE.

### A.4 DISTRIBUTION OF "PERSON" REPRESENTATIONS ACROSS THE BRAIN FOR ALL SUBJECTS

Figure S.12: **Presence of people in captions for S1-S8.** (a) Flatmap of cortex. (b) Inflated map of cortex. Dotted orange oval shows approximate location of TPJ, which is linked to theory of mind; green circle shows location of PCV, associated with third-person perspective of social interactions. For S5 alone we additionally label the mTL-bodies area.

| | Non-Person | | | | | Person | | Other | |
|------|------|------|------|------|------|------|------|------|------|
| | RSC | OPA | PPA | Food | Word | EBA | FFA | PCV | TPJ |
| S1 | 12.9 | 17.3 | 10.6 | 11.5 | 32.0 | 87.2 | 88.5 | 89.7 | 92.1 |
| S2 | 5.58 | 8.15 | 2.70 | 20.0 | 34.8 | 81.4 | 87.2 | 70.8 | 89.1 |
| S3 | 6.57 | 16.9 | 4.49 | 24.4 | 33.9 | 84.7 | 90.3 | 75.3 | 83.2 |
| S4 | 4.40 | 14.7 | 4.47 | 20.0 | 37.8 | 78.9 | 90.3 | 66.5 | 88.9 |
| S5 | 9.31 | 6.43 | 1.95 | 17.8 | 38.4 | 79.5 | 89.4 | 78.5 | 79.9 |
| S6 | 16.7 | 28.2 | 6.93 | 27.1 | 48.8 | 91.8 | 97.8 | 75.4 | 79.1 |
| S7 | 7.14 | 9.87 | 5.99 | 10.7 | 36.9 | 84.3 | 89.5 | 84.2 | 90.3 |
| S8 | 15.7 | 30.9 | 9.84 | 42.6 | 57.5 | 86.7 | 96.2 | 71.2 | 89.7 |
| Mean | 9.78 | 16.5 | 5.86 | 21.8 | 40.0 | 84.3 | 91.2 | 76.4 | 86.5 |

Table S.1: **Percentage of captions in each region that contain people for S1-S8.** We observe a sharp difference between non-person regions (Scene RSC/OPA/PPA, Food, Word), and regions that are believed to be person selective (body EBA, face FFA). We also observe extremely high person density in PCV — a region involved in third-person social interactions, and TPJ — a region involved in social self-other distinction.

## A.5 Additional extrastriate body area (EBA) clustering results

Figure S.13: **EBA clustering for S1/S2/S5/S7. (a)** EBA clusters. **(b)** Voxels which mention just a single person and those that mention multiple people. **(c)** Top nouns. Note that clustering was performed jointly on S1/S2/S5/S7.

| | Single | | Multiple | |
| --- | --- | --- | --- | --- |
| | EBA-1 | EBA-2 | EBA-1 | EBA-2 |
| S1 | 21.8 | 68.5 | 78.2 | 31.5 |
| S2 | 31.5 | 69.5 | 68.6 | 30.5 |
| S5 | 28.8 | 75.2 | 71.2 | 24.8 |
| S7 | 29.0 | 63.8 | 71.0 | 36.2 |
| Mean | 27.8 | 69.3 | 72.3 | 30.8 |

Table S.2: **Distribution of single/multi-person voxels within each EBA cluster.** After parsing each voxel's caption, we compute the single/multi voxels as a percentage of all voxels in the cluster that mention "person" class. We observe that EBA cluster 1 (EBA-1) has a higher ratio of voxels that mention multiple people. This is reflected in both the visualization, the nouns, and the human study on ground truth top NSD images for each cluster.

A.6 HUMAN STUDY DETAILS

Ten subjects were recruited via prolific.co. These subjects are aged $20 \sim 48$; 2 asian, 2 black, 6 white; 5 men, 5 women. For each NSD subject (S1/S2/S5/S7), we select the top-100 images for each cluster as ranked by the real average fMRI response. Each of the 100 images were randomly split into 10 non-overlapping subgroups.

Questions were posed in two formats. In the first format, subjects were simultaneously presented with images from the two clusters, and select the set where an attribute was ***more prominent***, possible answers include cluster-1/cluster-2/same. The second format asked subjects to evaluate a set of image from a single cluster, and answer yes/no on if an attribute/object-type was ***present in most*** of the images.

For the human study results in section 4.4, a human evaluator would perform 40 comparisons, from 10 splits and the 4 NSD subjects; with 10 human evaluators per question. We collected 1600 total responses for the four questions in the main text.

For the human study results below, a human evaluator would perform 16 judgements, from 4 splits and the 4 NSD subjects; with 10 human evaluators per question; across the 2 clusters of images. We collected 2560 total responses for the eight questions below.

Due to space constraints, we present the single set attribute evaluation (second format described above) results here in the appendix. We divide the results into two tables for presentation purposes.

| Are most images... | social | | | | sports | | | | large-scale scene | | | | animals | | | |
|---|---|---|---|---|---|---|---|---|---|---|---|---|---|---|---|---|
| | S1 | S2 | S5 | S7 | S1 | S2 | S5 | S7 | S1 | S2 | S5 | S7 | S1 | S2 | S5 | S7 |
| EBA-1 | 88 | 80 | 85 | 85 | 90 | 85 | 88 | 100 | 80 | 85 | 83 | 85 | 20 | 20 | 18 | 23 |
| EBA-2 | 28 | 23 | 35 | 45 | 28 | 25 | 30 | 50 | 38 | 28 | 33 | 60 | 30 | 30 | 30 | 28 |

Table S.3: **Human study on EBA clustering, first set of image attributes.** Each human study subject was asked to evaluate groups of 10 images, and answer yes/no on if an attribute was present in most images. Units are in %.

| Are most images... | artificial objs | | | | body parts | | | | human faces | | | | multi person | | | |
|---|---|---|---|---|---|---|---|---|---|---|---|---|---|---|---|---|
| | S1 | S2 | S5 | S7 | S1 | S2 | S5 | S7 | S1 | S2 | S5 | S7 | S1 | S2 | S5 | S7 |
| EBA-1 | 78 | 78 | 80 | 75 | 73 | 80 | 78 | 83 | 85 | 78 | 75 | 75 | 100 | 60 | 100 | 85 |
| EBA-2 | 85 | 75 | 83 | 80 | 35 | 30 | 40 | 55 | 28 | 20 | 15 | 45 | 23 | 8 | 18 | 25 |

Table S.4: **Human study on EBA clustering, second set of image attributes.** Each human study subject was asked to evaluate groups of 10 images, and answer yes/no on if an attribute was present in most images. Units are in %.

## A.7 TRAINING AND INFERENCE DETAILS

We perform our experiments on a mixture of Nvidia V100 (16GB and 32GB variants), 4090, and 2080 Ti cards. Network training code was implemented using pytorch. Generating one caption for every voxel in higher visual cortex ($20,000+$ voxels) in a single subject can be completed in less than an hour on a 4090. Compared to brainDiVE on the same V100 GPU type, caption based image synthesis with 50 diffusion steps can be done in $< 3$ seconds, compared to their gradient based approach of $25 \sim 30$ seconds.

For the encoder training, we use the Adam optimizer with decoupled weight decay set to $2e - 2$. Initial learning rate is set to $3e - 4$ and decays exponentially to $1.5e - 4$ over the 100 training epochs. We train each subject independently. The CLIP ViT-B/32 backbone is executed in half-precision (fp16) mode.

During training, we resize the image to $224 \times 224$. Images are augmented by randomly scaling the pixel values between $[0.95, 1.05]$, followed by normalization using CLIP image mean and variance. Prior to input to the network, the image is randomly offset by up to $4$ pixels along either axis, with the empty pixels filled in with edge padding. A small amount of normal noise with $\mu = 0, \sigma^2 = 0.05$ is independely added to each pixel.

During softmax projection, we set the temperature parameter to $1/150$. We observe higher cosine similarity between pre- and post- projection vectors with lower temperatures, but going even lower causes numerical issues. Captions are generated using beam search with a beam width of $5$. A set of 2 million images are used for the projection, and we repeat this with 5 sets. We select the best out of 5 by measuring the CLIP similarity between the caption and the fMRI weights using the original encoder. Sentences are converted to lower case, and further stripped of leading and trailing spaces for analysis.

## A.8    Top adjectives and more sentences

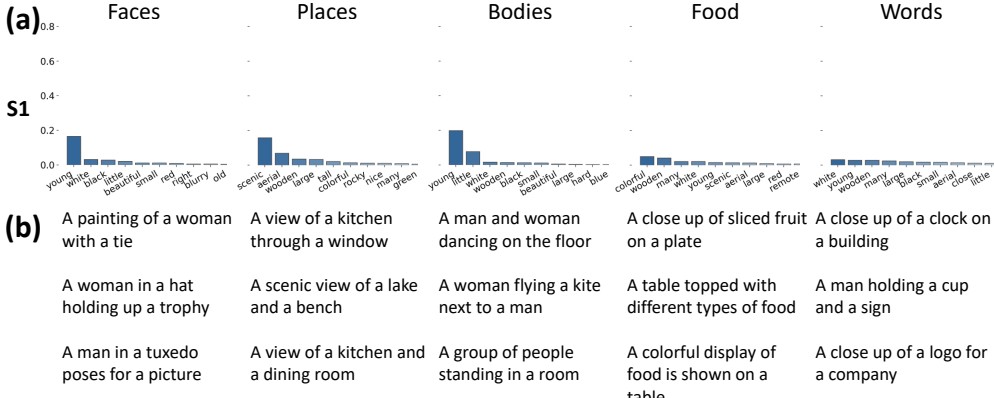

Figure S.14: **Top adjectives.** **(a)** We extract the most frequent adjectives in each category selective region, note how the adjectives are related to the semantic category in the brain. **(b)** Additional example sentences for each region. The category selective regions are identified via official NSD functional localizer experiments with a different stimulus set.

## A.9 ENCODER FITTING STABILITY

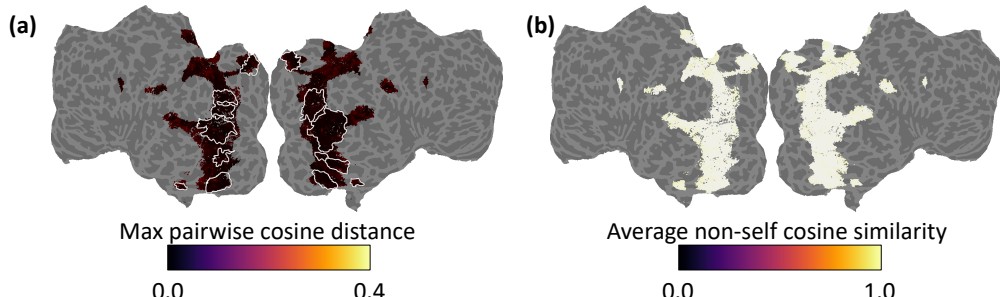

Figure S.15: **Result on 10-fold cross-validation. (a)** We measure the cosine distance of the voxel-wise weights across 10-folds. Visualized is the maximum any-pair voxel-wise distance. We find the average maximum any-pair across voxels is 0.02. **(b)** Average non-self pair-wise cosine similarity across the 10-folds. Note that for each fold, we randomly initialize the weights with kaiming uniform. We find that the fitting process is stable across repeats with an average non-self cosine similarity of 0.98.

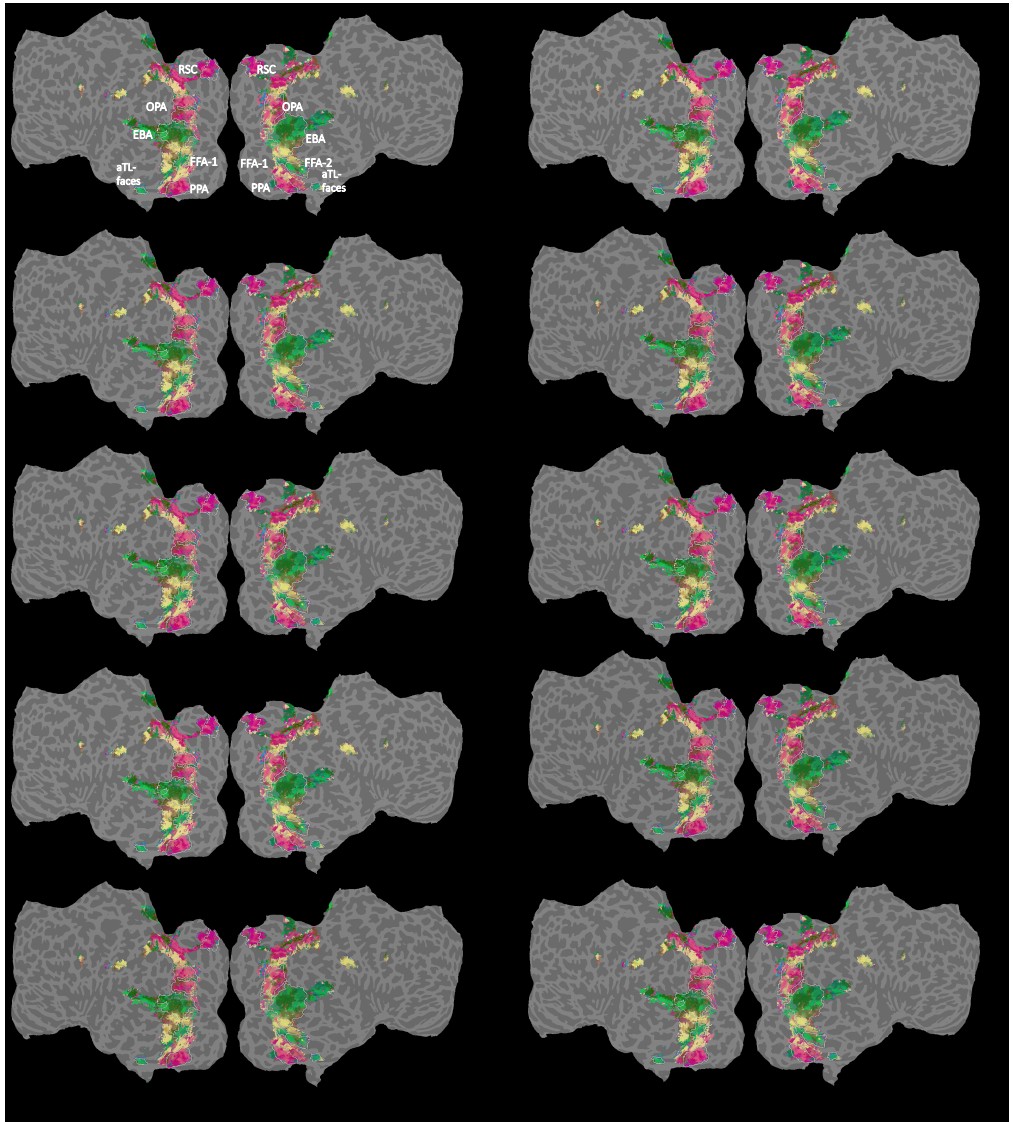

Figure S.16: **Projection of the 10-fold cross-validation encoders.** We perform UMAP projection using the basis from the main paper on each of the 10 encoder weights. We find that aside from minor differences in the FFA/food intersection on the right hemisphere, the large-scale distribution is similar.

A.10 GROUND TRUTH FUNCTIONAL LOCALIZER CATEGORY DISTRIBUTION

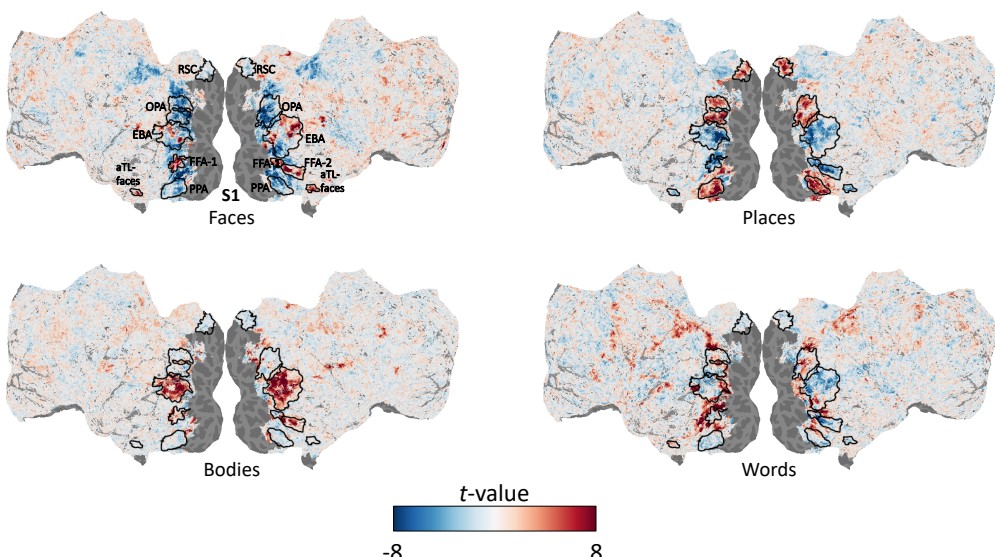

Figure S.17: **Ground truth $t$-statistic from functional localizer experiments.** We plot the ground truth functional localizer result $t$-statistics. The official functional localizer results are provided by NSD, and are collected using the Stanford VPNL fLoc dataset. Here red indicates a region which is activated by images from a category. This plot shows the broad category selectivity present in the high order visual areas.

## A.11 FINE-GRAINED CONCEPT DISTRIBUTION OUTSIDE EBA

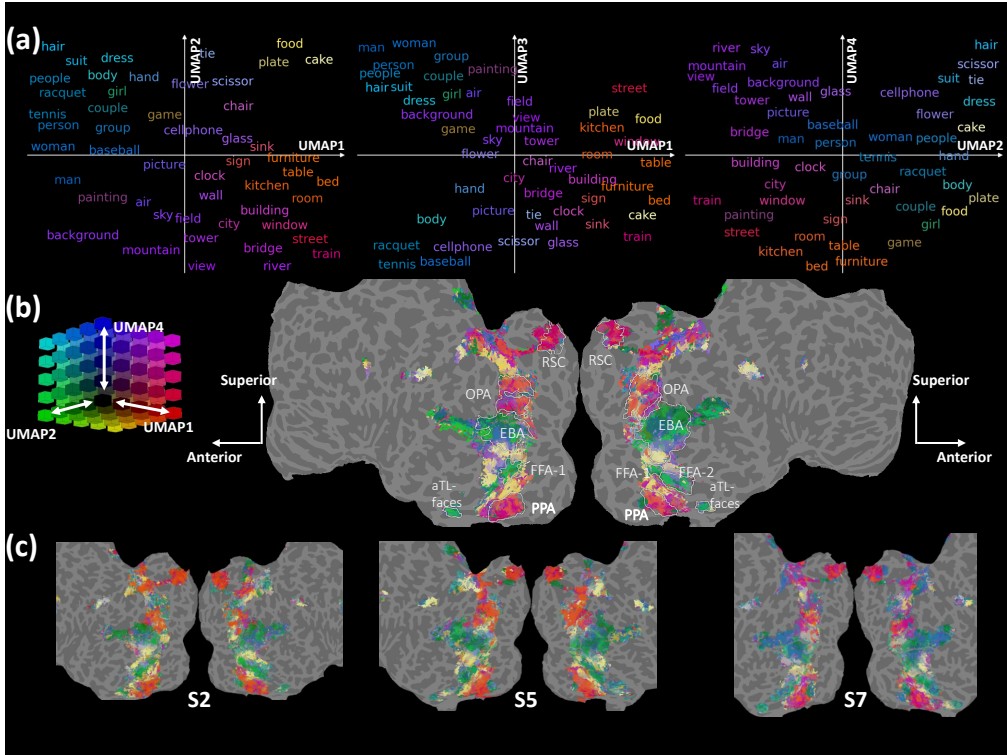

Figure S.18: **Additional UMAP visualizations.** Here we plot the UMAP dimensionality reduction, and identify the indoor/outdoor concept split in OPA using the 4th UMAP component. Note the Indoor (orange) and Outdoor (purple) gradient along the anterior-posterior axis.

A.12  NORM OF THE EMBEDDINGS WITH AND WITHOUT DECOUPLED PROJECTION

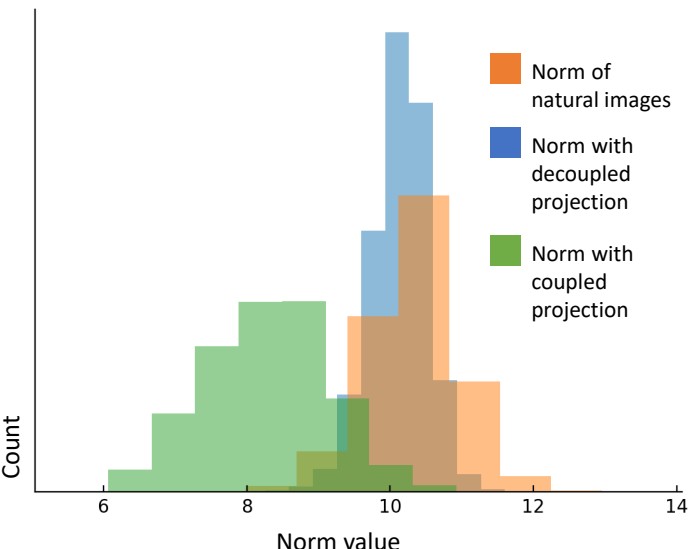

Figure S.19: **Norm of the embedding vectors.** In the main paper we decouple the projection of the norm and direction. Here we visualize the norm of natural image embeddings in orange, the norm of the post-projection weights using decoupled projection in blue, and the norm of the post-projection weights using coupled norm/direction projection in green. As vectors can cancel each other out, the use of decoupled projection in the main paper yields a better distribution alignment.

