# OpenReview forum: "BrainSCUBA: Fine-Grained Natural Language Captions of Visual Cortex Selectivity"
_ICLR.cc/2024/Conference — ICLR 2024 poster_

### Official Review · Reviewer_G8Co · 2023-11-01

**Soundness:** 3 good
**Presentation:** 4 excellent
**Contribution:** 3 good
**Rating:** 8
**Confidence:** 4

**Summary:**

The authors regressed fMRI neural activities onto CLIP embedding of images, after projecting the weight vector onto the “natural image distribution”, they use the projected weights to generate natural language description and image visualizations.  Using this method, they validated many functional properties of the human visual cortices and defined visual properties for some less charted lands on human cortex.

**Strengths:**

- This is a fast-paced field. The authors spend the time to add comprehensive references to previous literature, which form a strong foundation for evaluating this work.
- Simple but principled method. Noticing and addressing the modality gap with embedding projection seems like a key advance to make this work.
- The close comparison to BrainDiVE is interesting, which kind of suggests that the text caption captures the “essence” of high activation images, without the need for direct gradient ascent, i.e. the selectivity at least for the activation maximizing images is compressed in the words.
- The authors showed additional applications for neuroscience discovery which is super cool. This visualization/caption tool will help define the visual property of many uncharted lands. Evaluating the separation of clusters with human subjects is convincing.
- I can see this work’s approach is broadly applicable to multiple neuroscience modalities, e.g. ephys or imaging data from animals. Though language caption may not be a proper medium for them.

**Weaknesses:**

- Not much a weakness but more like a c**omment.** I think it seems common in this domain (e.g. neuron guided image generation), to pipe together several large-scale pre-trained multi-modal models with brain data and then train linear adaptors between them and then it will work.  So not quite sure how technical challenging this work is comparing to previous ones.

**Questions:**

### Questions

- In Eq. 4 why do you choose to use score to weighted average norm and direction separately, instead of averaging the vectors themselves? I can see arguments for both ways, but why do you choose this one?
- In Fig. 5, the Word Voxel visualizations using BrainSCUBA is always some round cake like object with text on it —— which is kind of strange, while the previous method (BrainDIVE) and NSD samples don’t have this feature. Where do you think this bias could come from?

---

> ### Author Response · Authors · 2023-11-16
> **Response to G8Co (1/2)**
>
> We appreciate the helpful and concrete suggestions by G8Co. We gladly welcome any questions and have provided our responses to your specific comments below.
>
> > **Q1) Difference between our work and (linear) decoding**
>
> We wanted to quickly summarize some conceptual differences between our work and the traditional decoding tasks -- which seek to reconstruct a stimulus from recorded brain activations. Recent decoding models indeed typically map from brain activation to a latent embedding using a linear model, and the result is coupled to a large image generative model.
>
> In contrast, our work seeks to understand the semantic selectivity of regions of the brain by generating natural language captions. Our work uses a linear probe which takes as input a CLIP image embedding and maps to voxel-wise brain activations. By constraining the encoding backbone to be unit-norm, the voxel-wise weight can be efficiently used to derive the optimal CLIP image embedding.
>
> A problem with this particular embedding is that is does not necessarily lie in the space of CLIP image embeddings for natural images. To that end, we propose an efficient parameter free method to project this embedding via a convex operation.
>
> Our method falls under the broad umbrella of work like [1,2,3,4,5,6]. These methods without exception rely on expensive gradient based optimization. An example includes BrainDiVE which requires **20~30 seconds per-sample**, compared to **2~3 seconds per-sample** for our method (including both caption and image generation) on the same hardware. This restricts these methods to analyzing broad swathes of the brain (typically ROIs on the order of hundreds to thousands of voxels).
>
> Our gradient free approach to selectivity analysis differentiates our work from previous gradient based models. **This huge speed improvement can facilitate efficient data-driven exploration of the broader visual cortex**, and we demonstrate this in section 4.3 **Figure 8**, where we perform voxel-wise caption generation and tagging, backed up by additional human studies on ground truth top images (1600 responses in **Table 3** and 2560 responses in **Table S.3/S.4**).
>
> [**1**] Inception loops discover what excites neurons most using deep predictive models (2019)
>
> [**2**] Neural population control via deep image synthesis(2019)
>
> [**3**] Computational models of category-selective brain regions enable high-throughput tests of selectivity (2021)
>
> [**4**] NeuroGen: Activation optimized image synthesis for discovery neuroscience (2022)
>
> [**5**] Brain Diffusion for Visual Exploration: Cortical Discovery using Large Scale Generative Models (2023)
>
> [**6**] Energy Guided Diffusion for Generating Neurally Exciting Images (2023)
>
> > **Q2) Decoupled norm and direction weighting**
>
> This is an excellent question that highlights a subtle aspect of our proposed method.
>
> Our goal in the projection step is to map the encoder weights to the domain of CLIP embedding of natural images. We decouple the norm and direction weighting because if we performed simultaneous weighted sum, the vectors could cancel out and end up with a different norm.
>
> Consider the following scenario, where we have 2 natural images, and their respective vectors are $a=[0.5, -0.5]$ and $b=[0.5, +0.5]$, both of norm ~$0.707$. Consider the case where the weights on a and b are $w_a=0.5$ and $w_b=0.5$, in this case, the resulting vector if we do not decouple norm/direction is $[0.5, 0]$ with norm $0.5$.
>
> Instead, if we perform decoupled norm and direction weighting, the result is $[0.5,0]/0.5 * 0.707 = [0.707,0]$, which would match the norm of the two input vectors.
>
> We further illustrate this effect in **Figure S.19 on page 38 of the revision**, we plot a histogram of the norms of 1 million natural CLIP image embeddings, the norms of encoder weights projected without decoupled norm, and the norms of encoder weights projected with decoupled norm. This figure shows that decoupled projection yields a better aligned norm distribution.
>
> We hope this clarifies our approach.

---

> ### Author Response · Authors · 2023-11-16
> **Response to G8Co (2/2)**
>
> > **Q3) Cake and food selectivity**
>
> BrainDiVE is guided by gradient signal of an encoder on a stochastic image diffusion model, while our approach works by generating captions conditioned on encoder model weights, which can be further used for diffusion image generation. Looking at the images in **Figure 5** and **Figure S.4 to S.11**, both BrainDiVE and our method generate a high frequency of cake-like objects, although our method indeed appears to have a higher frequency.
>
> We believe the difference primarily comes from the GPT based language model in one of two possible ways:
> 1. As BrainDiVE is not text-conditioned, it is free to generate images that are not quite recognizable objects, and yields supra-natural predicted activations even when evaluated with a second encoding model backbone. In contrast, our method typically yields coherent images with recognizable objects.
> 2. It is possible that a distribution difference exists in the language model itself, because cake is a more frequently mentioned food, and this is reflected in the captions, which is further reflected in the generated images.
>
> ⠀
>
> ⠀
>
>
> Thank you for your comments and suggestions! We welcome any additional questions and discussion.

---

### Official Review · Reviewer_79Ny · 2023-11-01

**Soundness:** 3 good
**Presentation:** 3 good
**Contribution:** 2 fair
**Rating:** 6
**Confidence:** 3

**Summary:**

The paper presents a data-driven method, BrainSCUBA, to generate natural language descriptions for images that are predicted to maximally activate individual voxels. This is achieved by first training a neural encoding model to predict the fMRI response from the image embeddings, then projecting the encoding weights into the representational space in a contrastive vision-language model, and finally producing interpretable captions for individual voxels. The results aligned with the previous findings about semantic selectivity in higher-order visual regions. The authors further demonstrated finer patterns that represent the "person" category in the brain.

**Strengths:**

- This method provides more interpretable neural encoding results.
- The generated text and synthesized images for individual voxels in the higher visual regions align well with previous research findings.

**Weaknesses:**

- The scientific contribution and significance of this work are unclear. The paper didn't provide much new insights into the neuroscience findings.
- The method of this paper is not sufficiently evaluated. There are multiple steps in the analysis, including training the encoding model, projecting weights, and transforming them into text or images, each of these steps (except encoding model accuracy) lacks a clear ground truth for comparison. Thus, we can hardly know how much the result deviated from the original voxel representation.

**Questions:**

- How does the encoding weight vector $W_i$ change regarding to the model's training procedure? For example, if different splits of training and testing datasets are used, to what extent does the $W_i$ fluctuate? This is concerned because all the following analyses depend on a robust and accurate $W_i$. And for a voxel that has poor prediction accuracy (e.g., correlation=0.2), we don't know how well $W_i$ can be trusted as the representative embedding of that voxel.
- The projection method is intuitive and smart, but there's no direct validation of its effectiveness. Is there a specific reason for using cosine similarity as the distance metric?
- The map in Fig.3 seems to be very categorical compared to the continuous and distributed brain map resulting from natural speech data (Huth et al. 2016). Does this imply the existence of more separable clusters in higher visual areas? Is there any finer organization within these clusters if you look at more UMAP dimensions, e.g., under the "human-related" group (blue voxels) or "scene-related" group (pink voxels)?
- The generated words in Fig. 4 mostly align with expectations. However, certain words, like "close" in the "Faces" and "Food" categories, are strange. Do the authors have an explanation for that?
- The findings in Fig.7 are interesting. However, I am concerned that the identified regions don't necessarily represent the concept of "people". A region that represents another semantic category (for example, "home") that often co-occurs with "people" words in natural language captions might also be highlighted with this method.

---

> ### Author Response · Authors · 2023-11-16
> **Response to 79Ny (1/2)**
>
> We thank Reviewer 79Ny for the detailed and constructive review! We look forward to further discussion, and are happy to answer any questions. We will address all questions below, and incorporate your suggestions into the paper.
>
> > **Q1) Neuroscience contributions**
>
> Our paper presents four key experiments, detailed in **sections 4.2, 4.3, and 4.4**, along with supplemental results. The first two experiments confirm existing knowledge (**4.2 and 4.3**), while the third and fourth (**4.4** and **supplemental**) offer **new insights into the neuroscience of person representation in the brain.** **Following your suggestion, we have clarified this in the revision**.
>
> There is no doubt that visual representations of the "person" class are extremely important in the human visual cortex, with representations of people emerging very early in infancy. In these two experiments, we seek to investigate the finer grained distribution of person representations in the brain.
>
> Using voxel-wise caption generation and part-of-speech tagging, we distinguish between person-related and non-person-related nouns. Our regions of interest (ROIs) were identified using NSD functional localizer experiments with an independent stimulus set.
>
> Experiment three visualizes the presence of the "person" class in different ROIs in **Figure 7** and **Table 2**, revealing a clear distinction between regions traditionally associated with person perception (like EBA and FFA) and those not (such as RSC/PPA/OPA). We also note increased selectivity in the word region, aligning with previous studies on the visual word form area (VWFA). Significantly, we find a high density of person representations in the precuneus visual area (PCV) and the temporoparietal junction (TPJ), as shown in **Figure 7** and quantitatively evaluated in **Table 2**, suggesting a role for these areas in processing people. The PCV has been implicated in first- versus third-person perspective judgements, while the TPJ has been implicated to play a role in theory of mind. **We believe we are the first to provide evidence for PCV and TPJ's role in person representations using a natural image viewing dataset, in contrast to experimenter-designed stimulus datasets.**
>
> Experiment four focuses on the extrastriate body area (EBA). We identify two distinct clusters in EBA related to single and multiple person representations, which have not been reported in past work. **A human study with 1600 responses (Tables 3) and 5120 responses (Table S.3 and S.4) supports our computational findings, indicating that EBA-1 is more selective for social images, while EBA-2 favors single person images. We believe this is a novel neuroscientific insight that has not been observed previously.**
>
> These findings, particularly regarding the EBA, offer novel insights and lay groundwork for future hypothesis-driven neuroscience research.
>
> > **Q2) Evaluation setup**
>
> Thank you for you suggestion!
>
> In the paper we perform the following validations:
> * In the encoder training step, we evaluate the $R^2$ on a **held out test set** which offers evidence that our encoder has strong performance. This test set is the ~$1000$ images viewed by all subjects. We find that we can achieve high explained variance across the higher order visual areas comparable to other works [1] using the NSD dataset, reaching 0.75 $R^2$.
> * In the projection step, we visualize the cosine similarity of the pre- and post-projection embeddings, and use this result to select the softmax temperature vector where cosine similarity plateaus in **Figure 2**. **In response to your suggestion, we have further added the standard deviation of cosine similarity in Figure 2 of the revision**
>
> We would like to further clarify how we perform evaluation of our entire system end-to-end. Key to our evaluation is the use of ground truth region of interest (ROI) masks. These masks ***are not*** derived from the stimulus set (10000 images), and instead they are derived from a functional localizer task. This task uses images with the presence of a semantic category (faces, places, words, bodies) using the Stanford VPNL fLoc dataset. This yields ground truth ROIs.
> We evaluate against these ROIs in a few ways:
> 1. UMAP dimensionality reduction shows that we recover the category selective regions. **In response to your suggestion, we have added visualization of the ground truth t-statistics for the fLoc experiments in Figure S.17 on page 36** This provides a further point of reference to compare **Figure 3** against.
> 2. We perform caption conditioned image synthesis for each ROI. Using CLIP for classification like BrainDiVE, we find our synthesized images have high semantic specificity matching the region's true semantic selectivity in **Table 1**.
> 3. (see next response)

---

> ### Author Response · Authors · 2023-11-16
> **Response to 79Ny (2/2)**
>
> 3. In the person experiments, we perform a human study and collect 4160 total responses (1600 in **Table 3**, 2560 in **Table S.3 and S.4**) on the ground truth top 1% fMRI images. The responses provide evidence that the caption analysis in **Figure 8b** and **8c** are valid.
>
> We hope this response helps clarify our evaluation setup.
>
> [**1**] Natural language supervision with a large and diverse dataset builds better models of human high-level visual cortex
>
> > **Q3) Stability of W***
>
> **In response to your suggestion, we have added two plots evaluating the stability of encoder weights in 10-fold cross validation in Figure S.15 in page 34** and the semantic stability of post-projection encoder weights in **Figure S.16 in page 35**. We find that our encoder is very stable across different training sets and initializations.
>
> > **Q4) Use of cosine similarity**
>
> To clarify, cosine similarity (dot product) is the officially recommended [text-image retrieval metric (link to github)](https://github.com/openai/CLIP/blob/main/README.md?plain=1#L117) for CLIP [2]. This is because CLIP embeddings naturally lie on the surface of the sphere.
>
> [**2**] Learning Transferable Visual Models From Natural Language Supervision
>
> > **Q4) Distribution of concepts in visual cortex**
>
> We are deeply inspired by the Huth et al. paper, and cite it explicitly in **Figure 3**. Huth et al. did in fact propose categorical regions with clear boundaries in Figure 3 of their 2016 paper using a cross-subject clustering-based approach.
>
> In addition, higher order *visual* regions are typically believed to have a clear categorical selectivity of voxels [3]. We further illustrate this in the revision by plotting the ground truth t-statistics from the NSD fLoc experiments in **Figure Figure S.17 on page 36** of the supplemental. The categories for fLoc were selected based on categories for which selective regions had been found as of 2015, with recent research from 2022 revealing an additional food-selective region.
>
> [**3**] The functional architecture of the ventral temporal cortex and its role in categorization
>
> > **Q4.1) Finer grained distribution of concepts**
>
> Beyond the representation of single people versus groups of people in extrastriate body area (EBA, blue voxels), there indeed exists a finer grained distribution of concept selectivity as revealed by UMAP dimension 4 in the occipital place area (OPA), which shows an indoor/outdoor split. This has been alluded to in BrainDiVE, and prior work on gradients in OPA [4], but to our knowledge this is the first time this has been independently confirmed using a language-based data-driven model. In the **revision we have added the OPA results in Figure S.18 on page 37**, using first, second, and fourth UMAP component as color components instead of the first three in **Figure 3**.
>
> [**4**] Processing of different spatial scales in the human brain
>
>
> > **Q5) "close" in part-of-speech tagging**
>
> This is due to the nature of the Spacy part-of-speech tagging engine, we checked the generated captions and "close" always occurs in the form of "A **close up** photo/picture of ...", where "close" is tagged as a noun. We believe this is reflective of how faces/food are depicted in our datasets.
>
> > **Q6) Co-occurrence statistics**
>
> We agree that identifying pure semantic selectivity is an important question. A significant advantage of our method compared to traditional functional localizer experiments is that our method processes naturalistic images, and does not need the images with a singular isolated concept.
>
> We perform a quantitative study of the density of "person" representation in different ROIs, and find that it is indeed much lower in voxels that do not code for person. Note for the word region has been demonstrated to exhibit person selectivity in prior work, in line with the higher person density shown in **Table 7**.
>
> We also extensively validate our results using a human study on the ground truth top fMRI images. In addition to the two-alternative forced choice human study to pick among clusters (**Table 3**, 1600 responses total), we also perform a human study where each attribute is judged to be present or not in a group of images (**Table S.3, S.4**, 2560 responses total). We believe this is a robust evaluation as it directly evaluates attributes. We find the human study provides additional evidence for the computational results in **Figure 8**.
> ⠀
>
> ⠀
>
> Thank you for the detailed and thoughtful feedback! We look forward to additional discussions. We hope that our clarifications have been insightful, and hope you could consider a more positive assessment of our work.

---

> > ### Comment · Reviewer_79Ny · 2023-11-18
> >
> > I thank the authors for their detailed response to my questions. I think most of my concerns have been addressed. I will raise my rating.

---

> > > ### Author Response · Authors · 2023-11-18
> > > **Thank you!**
> > >
> > > We appreciate the detailed suggestions and comments.
> > >
> > > Thank you again for the positive evaluation of this paper!

---

### Official Review · Reviewer_xxME · 2023-11-01

**Soundness:** 4 excellent
**Presentation:** 4 excellent
**Contribution:** 4 excellent
**Rating:** 8
**Confidence:** 4

**Summary:**

This paper proposes a method to find natural language description that maximally activates fMRI response of a voxel  to assess semantic-selectivity of different brain regions.

Key idea is to first train a linear encoder to predict voxel activations from a pretrained CLIP model. Then  Voxel weights are projected to CLIP embedding space to generate captions from a pretrained text decoder (CLIPCap).

The proposed approach is validated by comparing the selectivity obtained by the proposed method with brain regions which show selectivity for places, food, bodies, words and faces. The authors perform additional analysis to find a person specific region in body-selective areas demonstrating new scientific discovery from this approach.

**Strengths:**

1. Use of pre-trained modules (CLIP, CLIPCap) to generate captions which maximize a voxel’s response (Section 3, Figure 1)
2. Confirmation of BrainSCUBA’s findings on well-known category selective brain regions (Figure 4,5)
3. Demonstration of category selectivity through a text to image generation model. Figure 5 and Figure 6 show how this method can generate maximally activating images 10 times faster than gradient based maximization (BrainDIVE). The images generated using BrainSCUBA are also more human-interpretable as compared to BrainDIVE.
4. Finding person and non-person specific cluster within EBA  (Table 3, Figure 8)
5. Overall the paper is well written and easy to follow. The approach is presented in a simple yet comprehensive manner thus making it easier for new readers to follow.
6. The approach is validated both qualitatively (figure 5) and quantitatively using different metrics (Figure 6, Table 1,2)
7. The approach has potential to be extended to investigating more complex semantic specificity which are not feasible using category selective images only.

**Weaknesses:**

1. Minor: In Figure 3, color code used is not shown in the legend but is there in text in page 6. I recommend to add legend in the Figure also for clarity.
2. I believe this paper does not fully leverage the potential of BrainSCUBA. The captions generated are currently restricted to Nouns. Semantic selectivity using images is limiting as we have to find natural images that consist of only one concept without confounds. BrainSCUBA can allow a deeper investigation of social interaction through verbs , subject-object pairs and finding which regions are selective for specific interactions/emotions. I emphasize that I mentioned this in the weakness section as I would have loved to see more new results of semantic specificity (other than confirmations).

**Questions:**

1. Is the semantic selectivity found here limited by training set (NSD)? Can we expect to see semantic specificity that is present in CLIP training data but not in NSD images?
2. What was the intuition behind the choice  “A photo of a/an [NOUN]”. Did you consider other prompts ?

---

> ### Author Response · Authors · 2023-11-16
> **Response to xxME**
>
> We thank the Reviewer xxME for the helpful review and the positive evaluation of our work. We address your suggestions below, and update the revision to provide further clarification.
>
> > **Q1) Figure 3 clarity**
>
> We appreciate the suggestion! Indeed, we should provide additional context in the caption of Figure 3 using the color code.
>
> To clarify, the color code is derived from data without explicit access to the region of interest (ROI) masks. The purpose of **Figure 3** is to show that our method can recover the semantic selectivity across the broader higher order visual regions by aligning the embedding of a brain encoder (which uses a contrastive vision-language CLIP image backbone) with text embeddings from a CLIP text encoder.
>
> In **Figure 3**, the trace of ROIs are derived from an independent functional localizer task (stanford VPNL fLoc) provided in NSD for different semantic categories, including faces (FFA/aTL-faces)/places (RSC/OPA/PPA)/bodies (EBA).
>
> **We have clarified the ROIs using the color codes in the revision following your suggestions.**
>
> > **Q2) Potential of BrainSCUBA**
>
> We want to clarify that our system generates full natural language captions. These captions include nouns, adjectives, and subject/object interactions. We include examples of these captions in **Figure 1c**.
>
> Our purpose for focusing on noun based analyses in experiment 1 **(section 4.2)** and experiment 3 **(section 4.4 Table 7)** is primarily because nouns correspond to semantic categories, and enables a "sanity check" on the broader results given the broad consensus on category selectivity in higher order visual areas.
>
> Indeed our UMAP results, show that our methods can independently recover regions of food selectivity (yellow) corresponding to the food area (discovered in 2022 by three groups independently in [1,2,3]) in humans without using food specific stimulus.
>
> We further analyze the occurrence frequency of multiple-people and a single person in Figure 8B, and perform human behavioral study in **Table 3 and Supplemental section A.6** to show that our method corresponds well with human perceptual judgements. **We believe this observation of disjoint semantic coding in the EBA region is a novel discovery.**
>
> In response to your suggestion, **we have added additional plots of adjective frequency and example sentences in Figure S.14 on page 33 of the revision**, extracted from the BrainSCUBA generated captions using the Spacy part-of-speech engine. We hope this provides additional insight into the power of our method.
>
> > **Q3) Limits of semantic selectivity**
>
> This is a very interesting question. NSD is the largest 7T fMRI visual dataset, with 10000 images shown to each of 8 subjects. However 10000 is probably small relative to human visual experience.
>
> In theory it is possible that the CLIP embeddings could reveal selectivity to concepts not present in the NSD stimulus set, even when the encoder is fit on only NSD images.
>
> Consider the case where a voxel is perfectly monotonic w.r.t. orthodromic distance in CLIP space, and that the CLIP embeddings of all natural images form a (geodesic) convex set. Given the monotonic condition, it is possible that an image on the boundary of this set is predicted to be highly activating, even when this image is not present in the stimulus set. However, the monotonicity assumption is very strong and not true in practice, as it basically assumes the human brain reacts to visual stimuli exactly the same as a CLIP neural network. The language model itself is also an area of potential improvement. So to summarize, while it is possible in theory, the assumptions needed are strong.
>
> We believe collection of larger fMRI datasets and use of more powerful language models would help reveal additional selectivity.
>
> > **Q3) Prompt format**
>
> Our prompts used for Figure 3 follow the practice established by [1] (see section **3.1.4** of their paper), which popularized the use of this prompt format for zero-shot ImageNet classification. Recent work by CuPL [2] has established more sophisticated prompt formats which leverage GPT-3 augmentation. The original format is more commonly used, and we stuck with it in our work.
>
> [**1**] Learning Transferable Visual Models From Natural Language Supervision
>
> [**2**] What does a platypus look like? Generating customized prompts for zero-shot image classification
>
> ⠀
> ⠀
>
> Thank you again for your wonderful suggestions, and we’re eager to hear your thoughts on our clarifications. Please let us know if you have any additional questions or comments!

---

> ### Comment · Reviewer_xxME · 2023-11-20
> **Thanks for clarification and new results**
>
> I would like to thank authors for detailed responses to feedback from all the reviewers. I appreciate the additional plots authors generated for adjectives. The results in the plot (Figure S14, page 33) show a strong bias of white , young adjective in faces,  bodies and word regions. While this is not a strong concern, this bias should be mentioned and I would like to read authors thoughts on following
> 1. Where does this bias come from e.g. datasets (MS-COCO, NSD) ; models (CLIPCap, CLIP) ; or participants (were all participants in NSD from certain ethnicity); or other possible reasons?
> 2. How to address the above bias and other potential biases which may not be evident yet due to this approach not being fully explored?

---

> ### Author Response · Authors · 2023-11-20
> **Clarification on adjectives**
>
> Thank you for your suggestions!
>
> To clarify, we checked our generated captions, and were not able to identify any instance where "white" or "black" occurred in a racial sense. "White" and "black" primarily occurred in the form "A black and white photo of ...", and occurring otherwise as a generic non-racial color descriptor.
>
> In the food region, "white" primarily occurs in the form "... on a white plate".
>
> That said, we agree with you that there exist a racial imbalance in MS-COCO. And in such datasets, the percentage of people with a lighter skin color is very over-represented [1]. We believe the frequency of "young" likely also comes from caption data.
>
> Such a problem is also propagated to the diffusion generated images (Figure **S.4** to **S.11**). In the absence of racial imbalance the captions themselves, we believe the diffusion image bias can come from two sources:
> * The CLIP based caption embedding network (trained on a dataset, then frozen for use in diffusion)
> * The images used to train the cross-attention and diffusion model
>
> In particular, from private correspondence with contributors to Stable Diffusion, we understand that the frozen CLIP embedding model can have a large influence on the final diffusion output, even when the dataset for training the diffusion model is fixed. As an example, we were told that OpenAI's CLIP allows the diffusion model to better model artist styles, while Open-CLIP leads to a relatively poor ability to mimic artist styles, despite the larger CLIP training set used by Open-CLIP. And this happened when the Diffusion model was trained on the same dataset (but just swapped CLIP models). We understand this was one of the motivating reasons why Stable Diffusion XL moved to a dual embedding model that combines OpenAI CLIP with Open-CLIP.
>
> In addition, the diffusion model itself is trained on [LAION-Aesthetics, scroll down to browse example images](https://laion.ai/blog/laion-aesthetics/), which seems to also over-represent people with light skin.
>
> We believe one potential mitigation is to adopt weakly constrained captioning models such as ZeroCap [2], which score poorly on objective measures, but generate very diverse captions. Recent work has also explicitly proposed ways to balance racial and gender diversity of diffusion model output [3].
>
> **We have added additional discussion to the ethics section** on potential sources of dataset bias. Thank you for your suggestions! Please let us know if you have any additional comments.
>
> [**1**] Understanding and Evaluating Racial Biases in Image Captioning
>
> [**2**] ZeroCap: Zero-Shot Image-to-Text Generation for Visual-Semantic Arithmetic
>
> [**3**] Unified Concept Editing in Diffusion Models

---

> ### Comment · Reviewer_xxME · 2023-11-21
>
> Thanks for considering the suggestion to include a discussion on biases. I have no more concerns.

---

### Official Review · Reviewer_aznk · 2023-11-01

**Soundness:** 3 good
**Presentation:** 3 good
**Contribution:** 2 fair
**Rating:** 6
**Confidence:** 3

**Summary:**

This paper proposes to combine fMRI encoding models with pretrained vision-language models to find captions for different voxels in visual cortex. They use this method to perform exploratory data analysis on an existing fMRI data (Natural Scenes Dataset, NSD). They find that the method can reconstruct existing known distinctions in preferences in visual cortex (places, faces, bodies, food, words), and can uncover heretofore unrecognized distinctions in the extrastriate body area and in the social network of the brain.

**Strengths:**

Overall, this is an interesting application of pretrained vision-language models for better understanding how the brain is organized. The exploration of fine-grained distinctions in the social network of the brain (section 4.4) is quite convincing, especially given the human evaluation results. The paper is clearly written and the demonstration of the use case would, I believe, be of substantial interest to neuroscientists and cognitive scientists.

**Weaknesses:**

I don't believe that this submission is well-suited for a machine-learning-focused conference such as ICLR. It uses off-the-shelf, pretrained models to find information about visual cortex, which would be primarily of interest to neuroscientists and cognitive scientists. I cannot find substantial methodological advances here that would be of general interest to a conference aimed at machine learning researchers.

**Questions:**

My concerns are not about the soundness of the work–which is fine–but about the appropriateness for publication in a machine learning conference. I don't think that there is much extra work the authors could do to convince me otherwise. I'm open to re-evaluation of the paper's merits if the other reviewers deem it appropriate for this conference.

Nevertheless, I do have some minor comments:

Page 2: acitvations -> activations
Page 3-4: I found Figure 2b and its accompanying justification inscrutable. If the point is that there are no images close to unit sphere of captions, and hence blending (eq. 4) must be used to find something closer to the manifold of natural images, this does a poor job of conveying that, and text would be a better way of communicating that. If there is a different point they're trying to make, the authors should take a few sentences to explain what it is.
Page 7-8: I found the attempt at quantification conveyed in Figures 6 and Table 1 of dubious relevance. If the point of the method is to find visually coherent images that are easy for a human to understand the gist of, using sample images from the NSD itself would do just as well (e.g. Borowski et al. 2021). If the point is to get a better view of what an entire area is selective to, then it seems BrainDiVE works better. The authors should clearly state what they're trying to convey in these figures and tables, they take up a lot of space in the paper but don't add much, in my opinion.

---

> ### Author Response · Authors · 2023-11-16
> **Response to aznk (1/2)**
>
> We thank Reviewer aznk for the constructive review. We appreciate your recognition of the paper's technical soundness and potential interest to the fields of neuroscience and cognitive science - we address our paper's domain appropriateness for ICLR below. We also value your specific suggestions, which we have carefully considered in our responses below and in revisions to our paper.
>
> > **Q1) Suitability for ICLR**
>
> While we acknowledge your assessment regarding the suitability of our paper for ICLR, we argue that consistent with prior presentations at ICLR, BrainSCUBA contributes to the field by showcasing an *innovative application* of pre-trained vision-language models in field of neuroscience.
>
>
> Our paper was submitted under the **applications to neuroscience & cognitive science** primary area on OpenReview, one of the **primary areas officially recognized by ICLR 2024 in the [call for papers](https://iclr.cc/Conferences/2024/CallForPapers#:~:text=applications%20to%20neuroscience%20%26%20cognitive%20science)**. Since its inception in 2013, ICLR has included applications in neuroscience in the [topics of interest](https://iclr.cc/archive/2013/call-for-papers.html). The increasing prominence of computational neuroscience and cognitive science at ICLR is evidenced by a growing body of neuroscience papers published at ICLR ([see this anonymous list of ICLR 2020~2023 papers](https://github.com/anon-0518/ICLR_neuro_papers/tree/main)).
>
>
> In 2023, the following applications in neuroscience paper appeared at ICLR as an "oral": Aligning Model and Macaque Inferior Temporal Cortex Representations Improves Model-to-Human Behavioral Alignment and Adversarial Robustness. It is important to note that this paper's primary contribution was in the field of computational neuroscience and, methodologically, relied on straightforward fine-tuning of a pre-trained model.
>
>
>
> Key contributions of our paper include:
> * By using an encoder trained on a large dataset, we recover a *less noisy* estimate of neural semantic alignment (see **Figure 5**). As the encoder is parameterized as a linear probe of a unit-norm embedding, we can derive the optimal CLIP embedding efficiently with a normalization operation, bypassing the slow gradient-based optimization used in BrainDiVE.
> * We demonstrate that these embeddings diverge from natural CLIP image embeddings using UMAP. This motivates us to use a parameter-free method for aligning encoder weights with natural embeddings and generating voxel-specific captions. We show that our method can yield high categorical specificity, high predicted activations, and yield novel insights.
> * Our work pioneers voxel-wise semantic explanations in the brain setting, moving beyond traditional region-based approaches like NeuroGen/BrainDiVE. We have adapted and applied vision-language models in a way that has not been done before, particularly in the context of understanding the human brain's visual cortex.
>
> We do note with appreciation your willingness to reconsider based on broader reviewer feedback. As the other reviewers have not disagreed with our paper's suitability for ICLR, and two reviewers are positive about our work's contribution, **we hope you will reconsider the alignment of our work with ICLR's commitment to interdisciplinary research and the explicit call for papers on "applications to neuroscience & cognitive science"**.
>
> > **Q2) Typos**
>
> Thank you, we have corrected the typos in the revision of our work.
>
> > **Q3) Clarification of Figure 2b**
>
> **Figure 2b** shows that even when both the encoder weights and CLIP Image embeddings are projected to the *same unit-sphere*, there still exists a gap between the two embeddings. This plot was specifically motivated by [1](NeurIPS 2022).
>
> We aim to be consistent with prior work on identifying CLIP Image & CLIP Text embedding gaps, and use the same technique -- performing UMAP on two sets of embeddings simultaneously -- in showing a brain voxel embedding & CLIP Image embedding gap.
>
> **Following your suggestion, we have further clarified our motivation in the revision text.**
>
> [**1**] Mind the Gap: Understanding the Modality Gap in Multi-modal Contrastive Representation Learning (NeurIPS 2022)

---

> ### Author Response · Authors · 2023-11-16
> **Response to aznk (2/2)**
>
> > **Q4) Clarification of Figure 6 and Table 1**
>
> Thank you for bringing to our attention the two papers by Borowski and Zimmermann (2021), we agree they are interesting papers **and have included them as a citation in our revision.**
>
> **In neuroscience, novel captions/images provide a purpose beyond top-images, as they can act as a basis for future hypothesis based studies.** This has been demonstrated in [2,3,4] which utilized computationally identified selectivity to perform additional fMRI studies.
>
> The two figures you reference seek to explore the performance tradeoffs in BrainSCUBA versus BrainDiVE in achieving a 10x speedup. A major postive of BrainSCUBA compared to BrainDiVE is the magnitude improvement in speed. BrainDiVE has extremely high computational costs due to the explicit gradient based optimization used. This poses an obstacle in using BrainDiVE for computational characterization of the entire visual cortex which contains tens of thousands of voxels.
>
> On the same hardware with a V100 GPU, BrainDiVE uses around **20~30 seconds per image** (Page 41 of their paper), while our approach uses **2~3 seconds per caption + image**. Note both times ignore the training of the encoder, which is amortized for all future use. This magnitude improvement in performance is desirable if you want to explore finer grained regions in the brain.
>
> **Figure 6** qualitatively shows despite the 10x speed improvement, when evaluated using a different brain encoder backbone with state of the art imagenet performance (EVA02-CLIP trained using masked image modeling + CLIP objective), our generated images still achieve very high predicted activations that are only slightly lower than BrainDiVE.
>
> **Table 1** quantifies the semantic specificity using the exact same metrics in BrainDiVE. This shows that on average our method yields only a 3.8% and 9.6% gap in semantic specificity in S2 and S5. **We have further clarified our reasoning in the revision.**
>
> [**2**] Computational models of category-selective brain regions enable high-throughput tests of selectivity
>
> [**3**] Human brain responses are modulated when exposed to optimized natural images or synthetically generated images
>
> [**4**] Selectivity for food in human ventral visual cortex
>
> ⠀
> ⠀
>
>
> We respect your expertise and perspective, and we appreciate the opportunity to address your concerns. We hope that you might be willing to consider a more positive evaluation of our work.

---

> > ### Comment · Reviewer_aznk · 2023-11-18
> >
> > I think the list of previous neuroscience publications is pretty convincing. I had looked up 2020-2021 and couldn't find many hits, but it looked like they ramped up in 2022–2023. Since the other reviewers think it fits, and I have no qualms about the science, I've bumped up the rating to a 6.

---

> > > ### Author Response · Authors · 2023-11-18
> > > **Thank you!**
> > >
> > > We are deeply grateful for the positive assessment you've given to our paper! Thank you again for your suggestions and comments.

---

### Author Response · Authors · 2023-11-16
**Author Response Clarifications (1/2)**

## Summary of our response and revision
We are grateful to all reviewers for their time and their constructive suggestions, which we agree will significantly improve the communication of our work.

We are very encouraged by reviewers’ positive evaluation on the quality of this work, "interesting application of pretrained vision-language models" (**aznk**), "validated both qualitatively and quantitatively" (**xxME**), "more interpretable neural encoding results" (**79Ny**), "applications for neuroscience discovery which is super cool" (**G8Co**).

## General clarifications
### 1. Scope and experiments
* We propose BrainSCUBA -- a gradient free method to probe functional specialization in the higher visual cortex by generating captions. This paper is submitted under "applications to neuroscience & cognitive science", an official primary area of ICLR 2024.
* We leverage the observation that under the widely used CLIP backbone + linear probe used in fMRI encoding models, the optimal CLIP embedding can be efficiently derived. We make the observation that brain encoder weights typically lie outside the distribution of CLIP embeddings of natural images.
* We propose to use a parameter free way to align the weights of an image-computable voxel-wise encoder with the distribution of natural images, so that we can leverage an existing CLIP conditioned image-captioning model without retraining.
* This method is 10 times faster than traditional gradient based approaches (BrainDiVE), and can be used to generate novel images, which are **critical to future hypothesis driven visual experiments**.


We evaluate our method in three different ways, and utilize the ground truth category selectivity of voxels, measured using a functional localizer, as part of that validation. We also offer new neuroscientific insights on the representation of people in EBA. To summarize our evaluation methods:
* First, we directly analyze the captions from the voxel-wise captioning, and show that our results are aligned with the known properties of the human visual cortex.
    * We perform UMAP dimensionality reduction on aligned embeddings of nouns and brain encoder weights, without access to the functional localizer results. We show that our computed selectivity not only aligns with regions present in the 2015 Stanford VPNL functional localizer (faces, places, bodies, words), but can also recover the food selective regions surrounding FFA discovered in 2022.
    * We perform voxel-wise caption generation, and find that the frequency of top nouns in each broad category selective region are reflective of the ground truth category.
* Second, we perform caption conditioned image synthesis using a diffusion model.
    * We visualize the images, we find that the images are visually semantically coherent. We further perform CLIP 5-way classification using natural language and a different CLIP model, and find that the semantic classification performance is similar on average to BrainDiVE.
    * We train a different encoder using a different backbone, and find that our generated images can achieve high predicted activations using this new encoder.
    * These two experiments provide a visually interpretable overview of the performance of BrainSCUBA, and motivate the utility of BrainSCUBA for future hypothesis driven experiments.
* Third, we provide an investigation on the distribution of "people" representations in the brain backed by extensive human studies, and offer novel insights.
    * We explore the voxel-wise distribution of "people" across the entire higher visual cortex. We find that indeed our method predicts low density in non-people related areas (Scene regions - OPA/PPA/RSC; Food region; Word regions), and high density in people related areas (Face - FFA; Body - EBA). We also observe high density of people in TPJ (a region linked to theory of mind) and PCV (a region linked to first/third person perception). We believe we are the first to provide evidence for PCV and TPJ’s role in person representations using a natural image viewing dataset, in contrast to experimenter-designed stimulus datasets.
    * We perform clustering of EBA results, and identify a fine-grained coding for single/multiple people that is reflected in the captions. We perform a human study and collect $4160$ total responses evaluating the ground truth top-images in each cluster, which provides further evidence for our novel observations.

The ROI masks are derived from functional localizer results or masks from the official NSD paper (faces, places, bodies, words, OFA, OPA, TPJ, PCV), or obtained from other authors directly (food).

---

> ### Author Response · Authors · 2023-11-16
> **Author Response Clarifications (2/2)**
>
> ## 2. New results in revision
> 1. We add an overview of top-adjectives for each of the broad category-selective regions and provide additional example captions generated by our system (Page 33, Figure S.14; **xxME**)
> 2. We add an experiment that explores the stability of W* in a 10-fold cross-validation setting (Page 34, Figure S.15; **79Ny**)
> 3. We add an experiment that explores the stability of the projection operator in a 10-fold cross-validation setting with UMAP dimensionality reduction (Page 35, Figure S.16; **79Ny**)
> 4. We add a visualization of the ground truth *t*-statistic from the functional localizer experiments. Note that we use the functional localizer identified regions as part of our evaluation setup (Page 36, Figure S.17; **79Ny**)
> 5. We add a visualization of an additional UMAP component, which reveals an indoor-outdoor split in the OPA place coding region (Page 37, Figure S.18; **79Ny**)
> 6. We add a visualization of the embedding norms for natural images, decoupled projection, and non-decoupled projection (Page 38, Figure S.19; **G8Co**)
>
> ## 3. Revisions to the text
> 1. We better motivate our use of Figure 2 directly in the text. (section 3.2, **aznk**)
> 2. We add additional citations to Borowski and Zimmermann, and discuss the relationship of their work in artificial neural networks to our work on the brain. (section 2; **aznk**)
> 3. We add clarifications in the paper to better motivate the usefulness of generating captions that can capture voxel-wise selectivity. In particular, captions/images are useful for future hypothesis driven visual experiments. (section 4.3; **aznk**)
> 4. We add information about the category selectivity of each ROI in the Figure 3 caption (**xxME**).
> 5. We add additional details of our human study to provide evidence for our novel observation of fine-grained person selectivity in EBA (page 31, section A.6; **79Ny**)
>
> ⠀
> ⠀
>
> We genuinely appreciate the suggestions, and believe our paper will be improved with your feedback. Please let us know if you have any additional questions or comments!

---

### Meta-Review · Area_Chair_rNNz · 2023-12-05

**Metareview:**

This paper presents a novel approach for combining encoding models of fMRI data with pre-trained vision-language models. The approach is used to perform exploratory data analysis on a publicly available fMRI dataset. The reviewers praised the method for being simple yet principled and there was general agreement that the reported results were interesting. There were a few comments brought up during the reviewing process which appeared to be well addressed during the rebuttal. This included a question about the suitability of the paper for ICLR since the paper is an application of well-established ML methods to neuroscience. Given the fact that there is an official track at ICLR for "applications to neuroscience & cognitive science" and given the fact that all the reviewers found the paper sound and interesting, the AC recommends the paper be accepted.

**Justification For Why Not Higher Score:**

The ML methods described are well established, and the key contribution is in the application of these methods to novel problems (neuroscience).

**Justification For Why Not Lower Score:**

There is sufficient support from the reviewers, and "applications to neuroscience & cognitive science" is an official primary area of ICLR 2024.

---

### Decision · Program_Chairs · 2024-01-16

Accept (poster)